# Integrating taxonomic signals from MAGs and contigs improves read annotation and taxonomic profiling of metagenomes

Ernestina Hauptfeld [1], Nikolaos Pappas [1], Sandra van Iwaarden[1], Basten L. Snoek [1], Andrea Aldas-Vargas [2], Bas E. Dutilh [1,3,5] & F. A. Bastiaan von Meijenfeldt [1,4,5]

Metagenomic analysis typically includes read-based taxonomic profiling, assembly, and binning of metagenome-assembled genomes (MAGs). Here we integrate these steps in Read Annotation Tool (RAT), which uses robust taxonomic signals from MAGs and contigs to enhance read annotation. RAT reconstructs taxonomic profiles with high precision and sensitivity, outperforming other state-of-the-art tools. In high-diversity groundwater samples, RAT annotates a large fraction of the metagenomic reads, calling novel taxa at the appropriate, sometimes high taxonomic ranks. Thus, RAT integrative profiling provides an accurate and comprehensive view of the microbiome from shotgun metagenomics data. The package of Contig Annotation Tool (CAT), Bin Annotation Tool (BAT), and RAT is available at https://github.com/MGXlab/CAT_pack (from CAT pack v6.0). The CAT pack now also supports Genome Taxonomy Database (GTDB) annotations.

Metagenomic shotgun sequencing provides a single platform for exploring both the composition and the functional potential of diverse microbial communities[1–5]. While functional profiling maximizes the usage of the shotgun data, taxonomic profiling of metagenomes may involve mapping reads to a reference database containing specific marker genes[6–10], in which case only a portion of the data is used, and function can only be coupled to taxonomy for those reads that contain the marker gene. Alternatively, taxonomy can be assigned to as much of the data as possible by querying reads to a full reference database[11–14]. Metagenomic profilers carry out direct homology searches in DNA[12], protein[13], or k-mer space[11,15,16], and the resulting taxonomic profiles have been used in large-scale studies to characterize microbial communities of the oceans[1], the global topsoil[2], and to estimate the niche range of known microbial taxa[17].

Taxonomic profiles that are based on direct queries of individual reads to full reference databases give a comprehensive view of a microbiome but often contain annotations (i.e., assignment of sequences to a certain taxon) that are spurious. Assigning taxonomy based on homology searches is challenging, particularly for relatively short reads: (i) some genomic regions are highly conserved across taxa, making it difficult to discriminate between them; (ii) microbes have high rates of horizontal gene transfer[18,19], so the best hit in the reference database might be from a different taxon; (iii) environmental microbiomes may contain many novel taxa without close representatives in the reference database, resulting in possible annotation to e.g. a genus or species when the organism only shares the same order[20,21]; (iv) known taxa may contain novel genomic regions, resulting in no annotation of reads covering that region or annotation to a more distant relative, and (v) reference databases contain mis-annotated sequences[22]. These challenges are especially pronounced when directly comparing individual reads. Except for data from recent long-read sequencing platforms[23], reads are short sequences that contain

[1]Theoretical Biology and Bioinformatics, Science for Life, Utrecht University, Padualaan 8, 3584 CH Utrecht, The Netherlands. [2]Environmental Technology, Wageningen University & Research, P.O. Box 17, 6700 EV Wageningen, The Netherlands. [3]Institute of Biodiversity, Faculty of Biological Sciences, Cluster of Excellence Balance of the Microverse, Friedrich Schiller University, Rosalind Franklin Strasse 1, 07743 Jena, Germany. [4]Present address: Department of Marine Microbiology and Biogeochemistry (MMB), NIOZ Royal Netherlands Institute for Sea Research, PO Box 59, 1790AB Den Burg, The Netherlands. [5]These authors contributed equally: Bas E. Dutilh, F. A. Bastiaan von Meijenfeldt. ✉e-mail: b.e.dutilh@uni-jena.de; bastiaanvonmeijenfeldt@gmail.com

limited taxonomic information, leading to reads derived from a single strain potentially being assigned to several different taxa. Thus, while comprehensive, taxonomic profiles based on read annotations are inherently noisy with spurious annotations and often inaccurate[20].

Over the past decade, best practices in shotgun metagenomics have been established, including reference database-independent (de novo) assembly[24,25] and binning of metagenome-assembled genomes (MAGs)[26,27]. The resulting contiguous sequences (contigs) and especially MAGs allow for accurate detection of novel taxa. Contigs and MAGs are significantly longer than the original short sequencing reads, the additional data allowing for more reliable taxonomic annotation, either by multiple homology searches[21,28] or phylogenetic placement[29]. Long sequence length mitigates the errors in annotation discussed earlier because multiple taxonomic signals can be integrated, as the confidence in the taxonomic annotation is highest in MAGs, followed by contigs and then reads. Thus, assembly-based annotation of MAGs and contigs comprise a current best practice for analyzing shotgun metagenomic datasets. The reliable taxonomic annotations, together with read-based coverage of the MAGs and contigs, can be used to estimate taxonomic profiles that have little noise and high explanatory power[4,30,31]. However, even though taxonomic annotation is more accurate for longer sequences, they often represent only part of the metagenomic data and therefore provide an incomplete picture of the microbiome since most of the data are explained by reads, followed by contigs and then MAGs. As de novo assembly and binning depend on sufficient coverage of the genome sequence, it may be expected that especially rare microorganisms will be missed when MAGs or contigs are assessed. For a robust taxonomic profile that also includes rare microorganisms, an annotation protocol that integrates both taxonomic information from long sequences where available and short reads where not may thus be desirable.

Here, we present a Read Annotation Tool (RAT), an annotation pipeline for metagenomic sequencing reads that integrates accurate annotation of contigs and MAGs derived from de novo assembly and binning and direct homology searches of the remaining unassembled reads. RAT assigns taxonomy to reads, making use of assembly-based profiling by associating reads to longer sequences when possible and assigning taxonomy according to the most reliable taxonomic signal it can find (first MAGs, next contigs, and last individual reads). Contigs and MAGs are taxonomically annotated with the previously published tools Contig Annotation Tool (CAT) and Bin Annotation Tool (BAT)[21], which provide robust annotation based on open reading frame (ORF) prediction and comparisons to a protein database[32–34]. Direct read annotation by DIAMOND is then used for those reads that cannot be associated with a contig or MAG to improve the sensitivity of the resulting taxonomic profile. We show that, by integrating taxonomic signals from MAGs, contigs, and reads, RAT provides more accurate read annotations and taxonomic profiles than other state-of-the-art tools and accurately characterizes groundwater microbiomes with many novel taxa.

## Results and discussion

Natural microbial communities consist of many different microorganisms that can be identified and characterized via their DNA with shotgun metagenomic sequencing. For an overview of all microorganisms and their relative abundances in a sample, a comprehensive approach is to obtain taxonomic annotations for as many of the sequencing reads as possible. The resulting taxonomic profile reflects the amount of DNA that was contributed by the community members to the sequencing machine, as opposed to their cell count or number of genome copies[35]. The accuracy of the taxonomic profile depends on the reliability of the taxonomic annotations. While contigs and MAGs can be more reliably annotated than individual reads, in most metagenomic datasets, not all reads are assembled into contigs, and not all contigs are binned into MAGs (Fig. 1). To address this trade-off

between annotation reliability and the fraction of data that can be explained in a metagenome, we developed Read Annotation Tool (RAT) (Fig. 1b, c).

RAT annotates contigs and MAGs with the previously published tools Contig Annotation Tool (CAT) and Bin Annotation Tool (BAT), respectively. CAT and BAT predict ORFs on these longer sequences with Prodigal and query them against a protein reference database with DIAMOND blastp[33]. The taxonomy of the sequence is assigned based on the combined taxonomic signal of the individual ORFs, selecting higher-ranking taxa in cases where many conflicting signals are present[21]. Default options for the reference database include the NCBI non-redundant protein database (nr)[36] and, in the latest RAT update, the non-redundant set of proteins of the representative genomes in the Genome Taxonomy Database (GTDB)[37]. Alternatively, any protein database with taxonomic annotations can be supplied by the user. Next, individual reads are mapped to the contigs with BWA-MEM, and each read inherits the taxonomic annotation with the highest reliability: the MAG annotation if the contig is binned and the contig annotation if it is unbinned. Finally, the remaining sequences (reads that do not map to a contig and contigs that cannot be annotated by CAT) are annotated individually by querying them directly against the protein database with DIAMOND blastx in default sensitivity mode[33]. Thus, by assigning reads to the taxonomic annotation with the highest reliability, RAT reconstructs a comprehensive taxonomic profile with high accuracy (Fig. 1c, Supplementary Fig. 1). The final step in which sequences are individually queried to the protein database is optional, and depending on whether this step is included, we distinguish two RAT modes: in -mc mode, RAT only uses the most reliable read annotations, which are based on MAGs and contigs with ORFs. In -mcr mode, RAT also uses the read and contig annotations with DIAMOND blastx, which will include more tentative annotations while representing more of the data.

We evaluated RAT's performance of read annotation and how well the final taxonomic profile represents the microbial community. First, we addressed the trade-off between annotation accuracy and the fraction of reads that can be annotated by the different steps in RAT, using 28 samples of simulated data from three different datasets in the second round of the Critical Assessment of Metagenome Interpretation (CAMI2) challenge[38]. Second, with the same datasets, we compared taxonomic profiles predicted by RAT to those predicted by other commonly used state-of-the-art profilers. Third, we assessed the performance of RAT and the best-performing other profiler on real metagenomes. To this end, we analyzed 18 samples from three groundwater monitoring wells, a relatively unexplored high-diversity environment that contains many novel taxa[39].

### Including taxonomic signals from MAGs and contigs improves read annotation

To evaluate how the integration of different taxonomic signals influences the annotation of individual reads, we annotated simulated metagenomic datasets from the second CAMI challenge[38] (CAMI2) with RAT. CAMI2 simulated well-characterized microbiomes of the mouse gut and microbiomes with more taxonomic novelty from marine and rhizosphere environments. The 28 samples contained between 78–381 species and included raw reads, gold standard assemblies (the best possible assembly of the sequencing reads in a sample), and genome sequences of these species. In our benchmarks, we used the gold standard assemblies as contig input. Each dataset of the CAMI2 challenge has different strengths for benchmarking. The mouse gut dataset only includes taxa from known species. However, as shown in Supplementary Fig. 2a, reads from the mouse gut dataset show high sequence divergence from taxa in known databases (as represented by the NCBI nucleotide (nt) database) due to simulation of sequencing errors, thereby posing a challenge for metagenomic profilers and effectively simulating unexplored environments. The

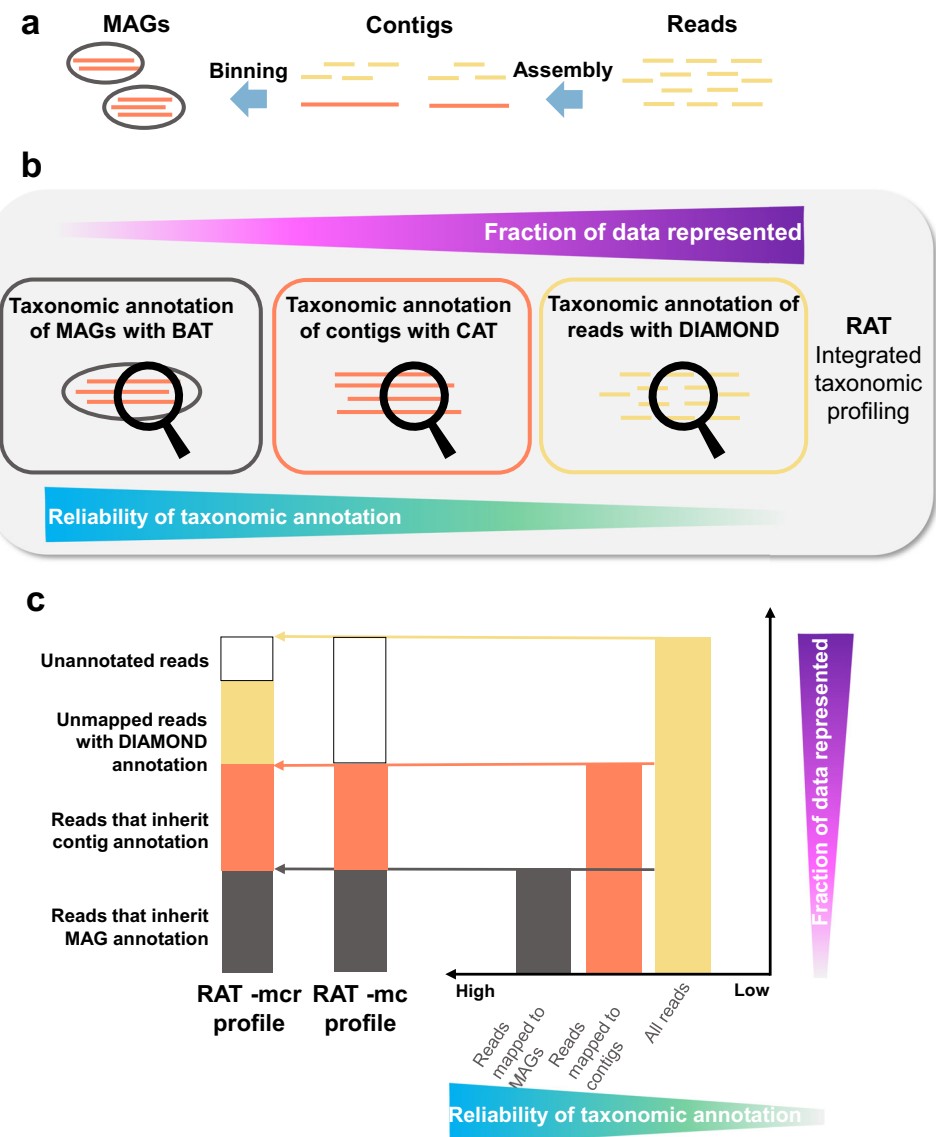

**Fig. 1 | The RAT workflow. a** Overview of a standard state-of-the-art metagenomics pipeline. **b** Overview of the RAT workflow: reads are mapped to contigs with BWA-MEM, which are binned into MAGs or unbinned. MAGs and contigs are taxonomically annotated using BAT and CAT, respectively. Unmapped reads and, thus far, unclassified contigs are annotated using DIAMOND. **c** Left: composition of an integrated taxonomic profile as reconstructed by RAT -mcr (for 'MAGs and contigs and reads', includes direct mapping of thus far unclassified contigs) and RAT -mc (for 'MAGs and contigs'). Right: schematic bar plot showing the fraction of the metagenome that can be annotated as reads, contigs, and MAGs.

rhizosphere dataset is the only dataset that contains eukaryotes, and its samples contain the highest fraction of reads belonging to unknown taxa, with up to 36.4% of reads not having a known species representative in nt (Supplementary Table 1). However, the rhizosphere samples have the highest fraction of reads mapping to MAGs, which is not representative of many biological samples, where reads are often unmapped or mapped to unbinned contigs (Supplementary Fig. 2b). Finally, the marine dataset contains 10.8–18.2% reads that belong to taxa without a known species representative in nt (Supplementary Fig. 2c). Conversely, reads from known taxa are highly similar to sequences in nt, making the samples representative of microbiomes that have been well-characterized (Supplementary Table 1, Supplementary Fig. 2a).

We compared five different methods for read annotation: (i) we annotated all reads directly with DIAMOND blastx, without mapping them to contigs or MAGs, (ii) RAT -cr for taxonomic annotations via contigs but ignoring MAG annotations, and direct read annotations for reads that did not map to contigs, (iii) RAT -mcr for annotations via MAGs, contigs, and reads, using the MAGs included in CAMI2 ('CAMI genomes'), (iv) RAT -mcr for annotations via MAGs, contigs, and reads, using MAGs binned by MetaBAT2 (ref. 26) (<10% contamination), and (v) RAT-mc for annotations via MAGs and contigs, using MetaBAT2 MAGs, but no direct read annotations (Fig. 2). Results were assessed at six taxonomic ranks (phylum, class, order, family, genus, and species) and we scored whether a read was correctly or incorrectly annotated, or unannotated (Fig. 2, Supplementary Fig. 3).

Direct annotation with DIAMOND blastx resulted in low accuracy at low taxonomic ranks like genus and species with a high fraction of mis-annotated reads (Fig. 2, Supplementary Fig. 3), revealing spurious annotations when mapping short sequences to a reference database. Accuracy in the mouse gut dataset is particularly low on species rank, with a true positive rate (TPR) of $14.1 \pm 4.5\%$ (mean $\pm$ standard deviation) (marine: $40.3 \pm 2.3\%$, rhizosphere: $25.0 \pm 20.6\%$) for DIAMOND. Despite using DIAMOND with the same reference database, RAT runs reduced mis-annotations and improved the fraction of correctly annotated reads at deep taxonomic ranks, highlighting the value of

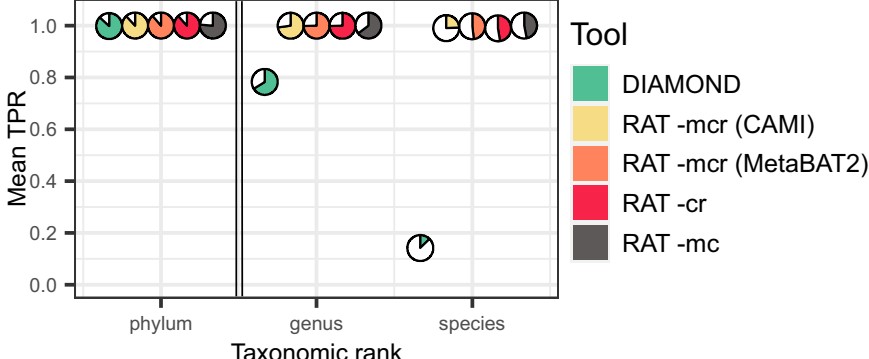

**Fig. 2 | Outcome of incorporating different taxonomic signals into read annotations on 10 samples of the CAMI2 mouse gut dataset.** 'DIAMOND' refers to using only direct read annotation in default sensitivity mode. 'RAT -mcr (CAMI)' refers to a RAT -mcr run (integrating MAGs, contigs, and reads) using the genomes that were provided by the CAMI2 challenge as MAG input. 'RAT -mcr (MetaBAT2)' refers to a RAT -mcr run with contigs binned by MetaBAT2. 'RAT -cr' refers to a RAT run without MAG input. 'RAT -mc' refers to a RAT -mc run, using only read annotation via mapping to MetaBAT2 MAGs and contigs, but no direct read annotation. The mean TPR refers to the fraction of correctly annotated reads per fraction of annotated reads averaged across the ten samples. The white section of the pie charts shows the fraction of unannotated reads. The same figure including results for all taxonomic ranks and for all benchmarked tools, as well as for the marine and rhizosphere datasets can be found in Supplementary Fig. 3. TPR true positive rate. Source data are provided as a Source Data file.

integrating information from taxonomically annotated MAGs and contigs (Fig. 2, supplementary Fig. 3).

When only taxonomic signals from contigs and direct read annotations are integrated (RAT -cr), the TPR increases compared to direct annotation with DIAMOND blastx, while the fraction of incorrectly annotated reads drops to 0.1–1% for all datasets. In addition, the fraction of reads with an annotation increases. This indicates that many previously mis- or unannotated reads are correctly annotated if they map to contigs (Fig. 2).

When taxonomic signals from both contigs and MAGs are integrated, the fraction of unclassified reads decreases compared to annotating without MAGs. In the CAMI2 mouse gut dataset, using the CAMI2 genomes as MAG input and binning the contigs with MetaBAT2 gave very similar results, indicating that current binning tools accurately group contigs from the same species together. Without using DIAMOND blastx to annotate the remaining unmapped reads and unclassified contigs (RAT -mc), the fraction of annotated reads decreases, while the true positive rate stays the same in the mouse gut dataset. In real biological datasets, RAT -mc is likely to annotate fewer reads on low ranks like species and genus. This is because higher diversity makes it more difficult to assemble reads into contigs than in the simulated CAMI2 samples, and in turn, fewer or shorter contigs lead to a smaller fraction of the reads being associated with longer sequences, which leads to less reliable annotations (see below, Supplementary Figs. 7 and 8).

In the marine and rhizosphere datasets, the same patterns are visible. RAT, on average, annotated the highest fraction of reads in the marine samples, followed by the mouse gut and rhizosphere dataset. All different tool settings showed a higher TPR on the marine dataset compared to the mouse gut or rhizosphere (Supplementary Fig. 3), likely due to the much higher similarity between the reads and the reference databases in the marine samples (Supplementary Table 1, Supplementary Fig. 2).

In conclusion, using the taxonomic signals from contigs and MAGs for read annotation leads to more reliable annotations than using direct querying of individual reads with DIAMOND.

### Including information from contigs and MAGs improves accuracy of taxonomic profiling

Metagenomics is used to analyze high-complexity microbial communities, including many different taxa with orders of magnitude of difference in their abundances. Taxonomic profilers aim to chart the community composition by estimating the relative abundance of all taxa in a sample. A good taxonomic profile contains as many members of the microbial community as possible while avoiding taxa that are not present in the sample. In practice, this often leads to a compromise between sensitivity (finding all taxa that are present and maybe some false positives) and precision (avoiding taxa that are not present and maybe some false negatives). To assess how the inclusion of contigs and MAGs affects the accuracy of taxonomic profiles, we used four metrics (sensitivity, precision, L1 distance, and weighted UniFrac distance) to compare the taxonomic profiles reconstructed by RAT and four state-of-the-art taxonomic profilers that carry out annotations via direct read mapping to the CAMI2 reference taxonomic profiles (Fig. 3). Centrifuge[12] compares read to the nucleotide database using a Burrows–Wheeler-transformation, Kaiju[13] annotates sequences in protein space, Kraken2 (ref. [11]) uses exact $k$-mer matches, and Bracken[40] uses the Kraken2 annotations for a Bayesian re-estimation of the abundances of taxa in the sample. The RAT output includes all taxa that are represented by at least a single read. However, as direct read annotations are known to be inaccurate, we limited the amount of noise by only considering taxa with a minimum abundance of 0.001% (which represents 4 reads in the CAMI2 samples) and applied this cut-off for all tools in the benchmark. This cut-off was applied separately on every taxonomic rank. Thus, even if a species with an abundance below 0.001% was removed, its reads could still be included at the genus rank if the genus was >0.001%.

In line with our first benchmark, the incorporation of taxonomic signals from MAGs led to more accurate profiles than using only taxonomic signals from contigs, as seen in the L1 distance and in the weighted UniFrac distance of RAT -rc and RAT -mcr. RAT -mcr slightly outperformed RAT -mc (Fig. 3, Supplementary Figs. 4, 5) in L1 distance and sensitivity, indicating that including direct read annotation leads to reconstructed profiles that are more similar to the reference profile than when relying solely on assembly-based profiling. Taxonomic profiles reconstructed by RAT consistently had lower L1 distances to the reference profiles than profiles reconstructed by Bracken, Centrifuge, and Kraken2 across all three CAMI2 challenge datasets (Fig. 3a, Supplementary Figs. 4 and 5). In comparison to taxonomic profiles reconstructed by Kaiju, RAT runs had slightly higher L1 distances on family, genus and species rank. Taxonomic profiles reconstructed by RAT had lower weighted UniFrac distances to the reference profiles than Bracken, Centrifuge, and Kraken2, except in the marine dataset (Fig. 3b, Supplementary Figs. 4, 5), while Kaiju performed similarly. The high performance of Bracken, Centrifuge, and Kraken2 on the marine dataset can likely be explained by the high similarity of the reads to the

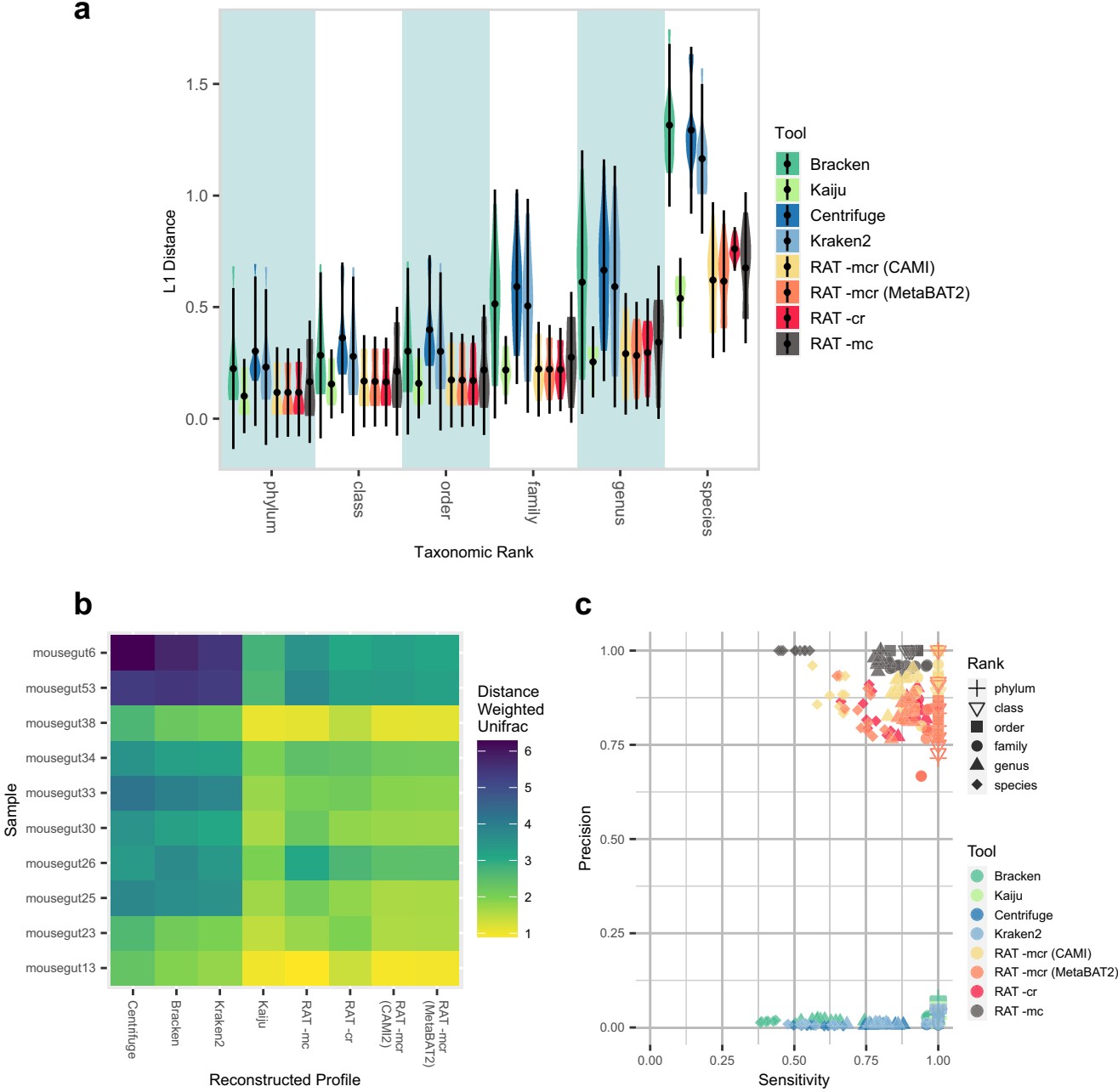

**Fig. 3 | Similarities between taxonomic profiles reconstructed by different tools and the reference taxonomic profiles of the CAMI2 mouse gut dataset using different similarity measures.** We only counted taxa as detected if their relative abundance was at least 0.001% (a minimum of 4 reads). **a** L1 distances between CAMI reference and reconstructed profiles by the tools in the study (mean ± 2 times the standard deviation, $n = 10$ samples). An L1 value of 0 means that the two profiles are identical; lower is better. The blue and white background shading facilitates differentiation between ranks and has no biological meaning. **b** Heatmap of weighted UniFrac distances between reconstructed and true profiles; a shorter distance is better. **c** Sensitivity vs. precision of the different tools. Different shapes signify the sensitivity/precision on different taxonomic ranks, different colors indicate tools; high precision + sensitivity is better. The same figures for the marine and rhizosphere datasets can be found in Supplementary Figs. 4 and 5. Source data are provided as a Source Data file.

sequences in the database reflecting well-characterized microbiomes. In well-characterized communities, annotations in nucleotide and $k$-mer space are put at an advantage compared to unexplored environments due to the high likelihood of long exact matches in the database. For communities containing mostly well-known organisms, tools such as Kraken2/Bracken or Centrifuge are therefore suitable. Conversely, more sensitive methods that annotate in protein space, especially ones that include a last-common-ancestor approach, such as both RAT and Kaiju, are at a relative disadvantage in well-characterized communities. As proteins are more conserved than nucleic acids, the annotations by these tools are more likely to be based on multiple taxa with the same

or similar amino acid sequences. This leads to a higher likelihood of missing annotations on low ranks compared to tools that search in nucleotide or $k$-mer space. Methods such as Kaiju or RAT are, therefore, particularly suitable for characterizing environments with organisms that exhibit high sequence divergence compared to their closest relatives in the databases.

RAT had a higher precision on all taxonomic ranks than the other evaluated tools (Fig. 3c). This means that RAT had fewer falsely detected taxa, in line with earlier observations of high precision of CAT and BAT annotations[21]. In the mouse gut dataset, RAT -mc maintained >0.94 precision on all taxonomic ranks, even when detected taxa were

**Table 1 | Runtime and memory usage of RAT and four other tools**

| | | mousegut6 | mousegut13 | Disk space output |
|---|---|---|---|---|
| Runtime | RAT -mc (robust) | 16 min | 16 min | 67 GB |
| | RAT -mcr (sensitive) | 106 min | 139 min | 79 GB |
| | Centrifuge | 11 min | 11 min | 900 MB |
| | Kaiju | 16 min | 15 min | 4.2 GB |
| | Kraken2/Bracken | 2 min | 2 min | 6 GB |
| Maximum memory usage | RAT -mc (robust) | 43 GB | 48 GB | |
| | RAT -mcr (sensitive) | 84.1 GB | 115.4 GB | |
| | Centrifuge | 240.4 GB | 240.6 GB | |
| | Kaiju | 127.4 GB | 127.4 GB | |
| | Kraken2/Bracken | 55.6 GB | 55.4 GB | |

Tools were run on two simulated datasets from the CAMI2 challenge (mouse gut sample 6: 33,098,456 reads, and sample 13: 33,184,772 reads). RAT -mc: annotation based only on contigs and MAGs, and RAT -mcr: annotation also based on DIAMOND annotation of unmapped reads. Kraken2 and Bracken are run together. All runs were performed using 16 CPU cores when the program allowed for multithreading (Intel Xeon Gold 6240R).

not limited by a minimum relative abundance cut-off (Supplementary Fig. 5). The same pattern can be seen in the marine (>0.93) and rhizosphere (>0.83) datasets, where RAT consistently showed higher precision than the other evaluated tools. Thus, like CAT and BAT on which its annotations are based, RAT -mc tends to avoid spurious annotations at low taxonomic ranks like genus and species in cases where conflicting taxonomic signals arise. For RAT -mcr, precision remained higher than that of the other evaluated tools across taxonomic ranks, but precision was lower than that of RAT -mc. The minimum relative abundance cut-off greatly improved the precision of RAT -mcr (cf. Fig. 3c and Supplementary Fig. 6). Spurious annotations are introduced when short sequencing reads are directly annotated in the direct annotation step of RAT and by the other evaluated tools. However, because of the prioritization of taxonomic signals in RAT, a smaller fraction of reads is annotated directly, leading to fewer spurious annotations in the first place. By setting an abundance cut-off (e.g., 0.001% of reads as in this benchmark), RAT can profit from the high sensitivity of the DIAMOND blastx step (finding taxa that might not be detected using just contig and MAG annotations) while further minimizing the number of falsely detected taxa (by excluding spurious annotations that have a very low abundance).

RAT's overall high precision can be explained by its integrated taxonomic profiling approach, which improves annotations in most of the challenges discussed above. Reads that map to conserved or horizontally transferred regions, or map to novel genomic regions of a known taxon, are likely to get the correct annotation with RAT because the surrounding regions of the genome are considered in the annotation via the contig and/or MAG. Reads belonging to novel taxa within known clades are also more likely to get correctly annotated, as when the reads are assembled into contigs or MAGs, RAT may annotate them on a higher taxonomic rank. For example, if the closest related sequences in the reference database are found among different species in a genus, the sequence will be annotated at the genus rank (see Methods). The difference in precision between the different approaches shows that reads that are annotated via direct read mapping instead of by being associated with a contig or MAG are far more likely to get falsely annotated. RAT's approach reduces the number of falsely detected taxa from 200–4000 by the other evaluated tools to between 0 (RAT -mc) and 38 (RAT -mcr).

All evaluated tools showed high sensitivity from phylum down to family rank, detecting most of the taxa that were present in the reference profiles (Fig. 3c, Supplementary Figs. 4, 5). This is consistent with increased barriers to horizontal gene transfer at higher taxonomic ranks[41]. Including direct read annotation consistently increased RAT's sensitivity compared to RAT -mc on all ranks and across all three datasets. One may expect that these classifications are less robust than those annotated via MAGs or contigs. All tools displayed the highest

sensitivity in the marine dataset and the lowest in the rhizosphere samples. RAT's high performance on the CAMI2 datasets is in part due to the fact that a large fraction of the reads map back to annotated contigs (mouse gut: 81.6 ± 6.6% (mean ± standard deviation), marine: 97.4 ± 0.2%, rhizosphere: 95 ± 4.7%) and MAGs (mouse gut: 75.8 ± 6.4%, marine: 90.5 ± 0.5%, rhizosphere: 91.7 ± 8.0%, Supplementary Fig. 7, supplementary Table 1). These numbers are often lower in real metagenomic datasets (see below). The result is that most reads are annotated in the most reliable MAG and contig annotation steps, and few reads are annotated directly with DIAMOND, reducing the probability of spurious annotations (Supplementary Figs. 7 and 8). To show the effect of using simulated vs. biological data, we also tested RAT on a set of 18 groundwater samples (see below).

**Usage, runtime, and memory requirements**

Next, we compared the runtime and memory requirement of RAT to the other tools on the mouse gut samples 6 and 13 (Table 1). RAT does not assemble and bin metagenomes but rather takes assembled contigs and associated MAGs as input from the user. Other user input includes the CAT database and taxonomy folders, as well as the sequencing reads. If a previous RAT run was interrupted, the intermediate files can be used as input to shorten runtime. If CAT and/or BAT have already been run on a dataset, the output files can also be used as input for RAT. Although assembly and contig binning can take hours or days to run (for example, the two mouse gut samples took around 2 h to assemble and bin, Supplementary Table 2), they are a common procedure in many metagenomics studies, as they provide valuable genomic context information to short sequencing reads with relatively little risk of generating chimeras[42].

Kraken2 was the fastest tool (01m49s), RAT --mcr was the slowest (02h05m10s), and all other tools, including RAT -mc performed the jobs in 16 min or less. In terms of memory usage, all tools can be run on a 256 Gb server. RAT -mcr had a higher memory footprint than Kraken2, but lower than Kaiju and Centrifuge. RAT -mcr varied in RAM and runtime between the two samples because it loads different amounts of unclassified reads and contigs into memory depending on the sample.

**The expanded CAT pack facilitates the detection and annotation of unknown microorganisms**

The simulated data provided by the CAMI2 challenge differs from real biological datasets. In the CAMI2 datasets, Illumina sequencing experiments were simulated of relatively low-diverse microbiomes containing mostly reads of known species (Supplementary Fig. 2). Annotations are facilitated by the fact that on average >80% of the reads mapped back to a MAG or contig from a gold-standard assembly, while in biological datasets, this percentage can be much lower

(Supplementary Figs. 4 and 5). In addition, particularly in microbiomes from under-studied environments, unknown lineages are often detected that are only distantly related to known taxa in reference databases. Awaiting taxonomic classification of these microorganisms, a higher-rank taxonomic annotation of the sequence at e.g., family or phylum rank may be appropriate in these cases.

RAT provides a framework for assessing these unknowns. Because reads are classified via CAT and BAT, annotations are made at the appropriate taxonomic rank. CAT and BAT assign individual ORFs to the last common ancestor of all hits that have a similar bit-score to the best hit and annotate the contig or MAG using a bit-score-based voting scheme that selects the taxon at which a certain fraction (in the RAT workflow, the majority) of the ORF assignments agree[21]. Novel sequences have many distinct hits and are thus only annotated at a high taxonomic rank, reflecting their unknownness. MAGs that only receive a high taxonomic rank annotation by BAT may be further investigated with phylogenomic software for strain-level resolution. Since the quality of RAT results is highly dependent on the quality of the input data, we recommend using high-quality assemblies and only including MAGs with low contamination (e.g. <10% contamination according to CheckM[43]). Contaminated MAGs can be mis-annotated or annotated at a high trivial taxonomic rank, in which case a contig annotation is more reliable. MAG completeness is less relevant for RAT, as MAGs with low completeness typically still include more than one contig from the same microorganism, creating a stronger taxonomic signal than present on the individual contigs.

To challenge RAT with real datasets, we selected relatively unexplored groundwater samples taken 12–64 m below the surface level from three different monitoring wells in a Dutch agricultural area, which we previously found had high microbial diversity and contained many novel taxa[39]. We performed a metagenomic analysis including quality control, assembly[24], and binning[26,27,44], which produced 514 MAGs. We supplied the reads, 2,770,251 contigs, and 423 medium- to high-quality MAGs (completeness ≥ 50%, contamination < 10%; see ref. [45]) to RAT to reconstruct taxonomic profiles of the groundwater samples, using nr as a reference database. In addition, the medium- to high-quality MAGs were dereplicated[46], and the resulting 195 representative MAGs were placed in a phylogenetic tree showing their relationships and abundance across samples (Supplementary Fig. 6).

RAT annotated 22.0 ± 8.7% (mean ± standard deviation) of reads by mapping them to MAGs, much less than in the simulated CAMI2 datasets (see Supplementary Table 1, supplementary Figs. 4 and 5), reflecting the high complexity of the groundwater samples. RAT classified 20.9 ± 3.2% of reads via unbinned contigs annotated by CAT and 0.35 ± 0.23% via contigs annotated by DIAMOND. Finally, DIAMOND blastx annotated an additional 23.0 ± 3.3% of the reads. These unmapped reads represent sequences with low coverage that could not be assembled into contigs, and based on the results with simulated data above, we expect to represent more spurious results.

The taxonomic profile reconstructed by RAT -mcr showed that most reads belonged to unclassified bacteria, including the phyla *Chloroflexi* and *Deltaproteobacteria* (Fig. 4a). *Chloroflexi* bacteria utilize a variety of electron acceptors, including oxidized nitrogen or sulfur compounds. A comparison of the 18 reconstructed taxonomic profiles showed that Sample 23-2 contained relatively many *Chloroflexi* reads, while the *Deltaproteobacteria* were rare. Although many of the microorganisms in this sample could only be classified on high taxonomic ranks, 22 MAGs from these phyla represented 31.1% of the reads in the sample (see Supplementary Fig. 6).

Next, we compared the taxonomic profiles of the groundwater metagenomes as predicted by RAT and Kaiju, as Kaiju was the best-performing other tool in the previous benchmark. Both tools classified two-thirds of the data (RAT: 68.9 ± 5.8% of reads, Kaiju: 63.8 ± 5.6% of reads, Supplementary Table 2). However, RAT classified these reads as belonging to roughly 20% of the taxa that Kaiju predicted (Fig. 4b).

Bearing in mind the high precision of RAT (Fig. 3), we propose that the taxa predicted by RAT are a more parsimonious interpretation of the metagenomic data than those predicted by Kaiju. To visualize the potential overestimation of taxa due to spurious annotations, we made rarefaction curves for the results of RAT -mcr, RAT -mc, and Kaiju. Without a minimum relative abundance cut-off, rarefaction curves of RAT -mcr and Kaiju results did not level off. This pattern was also observed in simulated data containing a known number of 110 species (Supplementary Fig. 10) and thus points to an overestimation of taxa richness. This reflects the spurious annotations of individual reads and indicates that, without a cut-off, deeper sequencing of the same sample would lead to higher predicted richness. The rarefaction curve of RAT -mc leveled off in the groundwater data, indicating robustness towards falsely detected taxa in the RAT -mc workflow. With a minimum relative abundance cut-off of 0.001%, all rarefaction curves leveled off, although the different tools predicted different taxa richness. Kaiju estimated a much higher richness than RAT in both -mc and -mcr mode (Fig. 4c). Combined with the RAT results on simulated data where RAT -mc underestimated richness while RAT -mcr included some false positives (Supplementary Fig. 10), this shows that: (i) RAT -mc is the best-suited RAT workflow in experiments where reliability is crucial, but it will likely not detect all of the rarer taxa, while (ii) RAT -mcr is more sensitive and will detect more taxa at the risk of including a few of them spuriously.

Since RAT annotates all reads in a metagenome, the resulting taxonomic profiles reflect sequence abundance as opposed to taxonomic abundance[35]. This means that RAT reports the abundance of a taxon as a fraction of total DNA in the sample rather than as the number of genome copies, which can, for example, be estimated by querying marker genes[6–8]. It may be expected that the resulting relative abundance profile is skewed towards microorganisms with larger genomes since they provide more DNA to the sequencing machine and thus contribute more reads than organisms with small genomes. To convert sequence abundance to genome copies, relative abundances have to be normalized by genome length, which is often unknown and can vary widely even between strains of the same species[47]. For novel microorganisms, genome sizes of closely related species might not be available. For these reasons, RAT, by default, does not convert sequence abundance to taxonomic abundance. However, the CAT pack provides a table with weighted mean genome sizes for most known bacterial and archaeal taxa at all ranks based on genomes deposited in the BV-BRC database[48]. This allows users to estimate the relative genome abundances from relative sequence abundances if they wish.

## GTDB compatibility provides lower-rank annotations on biological data

The performance of a taxonomic profiler can only be as good as the underlying database that is used to annotate the data. Further, the database used can only be as good as the taxonomy that it is based on. The taxonomy of living organisms is still regularly being updated[37,49]. Curated databases, such as GTDB[37] may provide better precision for profilers, but they might reduce sensitivity as they do not contain all known taxa. Conversely, comprehensive databases such as nr and nt[36] contain more sequences, increasing the sensitivity of the profiler but the taxonomic annotation of those sequences might be of lower quality, so precision might be sacrificed.

To provide the user with as much freedom as possible, the CAT pack is now compatible with the GTDB database as well as the nr database, which includes NCBI Taxonomy. Although the nr database is larger than GTDB (591,417,602 versus 250,802,978 protein sequences as of November 2023), GTDB's automated classification based on genome phylogeny makes the database more robust and less noisy than nr. CAT now includes automatic download, database preparation, and sequence annotation with CAT, BAT, and RAT based on GTDB. To

## a

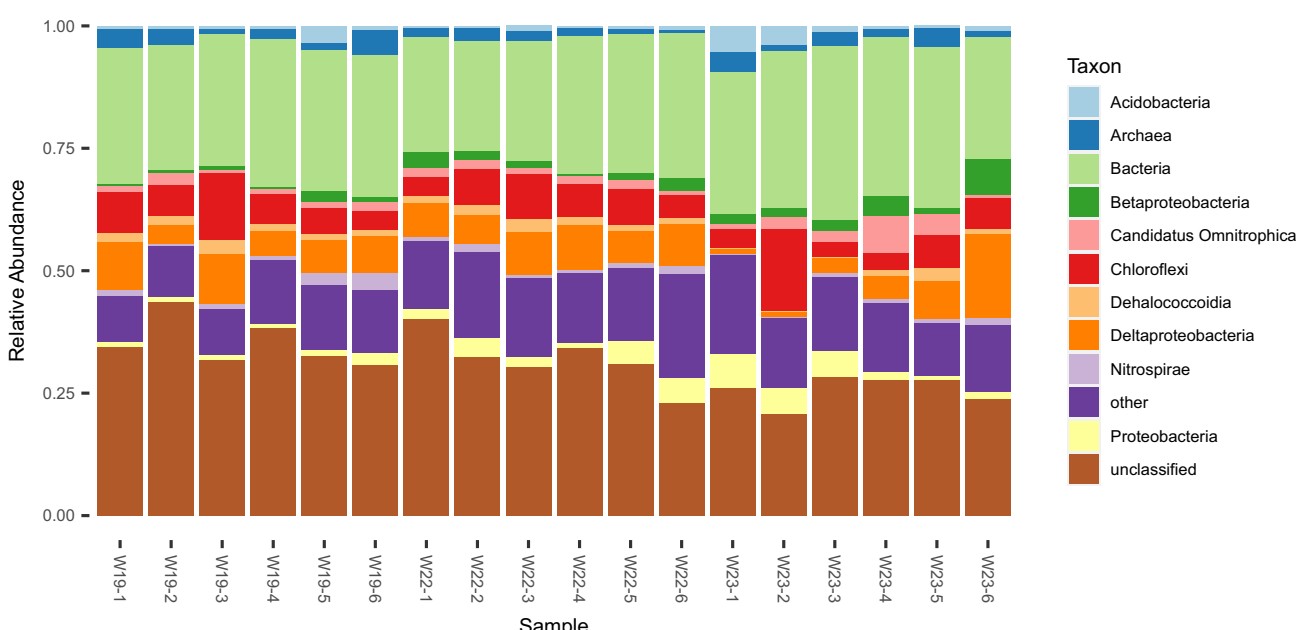

## b

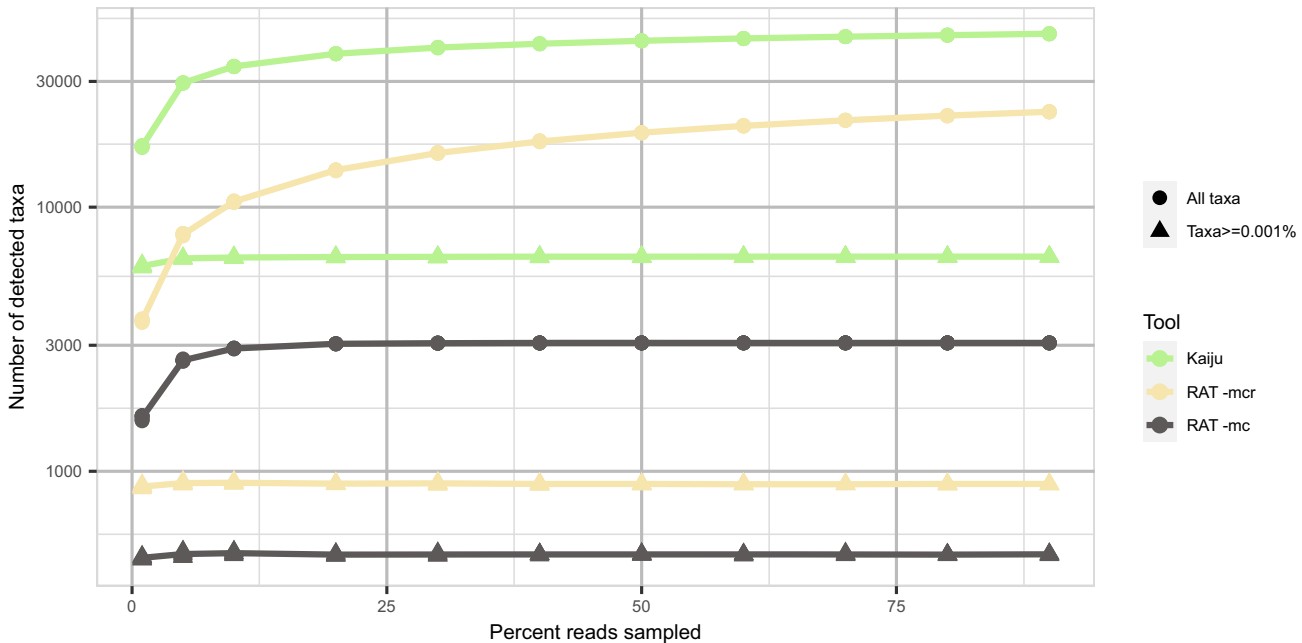

**Fig. 4 | Taxonomic profiling of groundwater metagenomes. a** Microbial profiles of groundwater samples on taxonomic rank class as reconstructed by RAT -mcr. **b** Rarefaction curves of the number of taxa detected in sample W23-2 by RAT -mc, RAT -mcr, and Kaiju. Triangles indicate the number of taxa detected in profiles when a minimum abundance is required to consider an organism as detected. Circles indicate the number of taxa detected without a cut-off. Source data are provided as a Source Data file.

show the difference in annotation between the two databases, we ran RAT on the groundwater samples described above with the nr data-base and with GTDB release 202 (Supplementary Figs. 9, 11, supplementary Table 5). In the MAG step, RAT annotated more MAGs with GTDB than with nr (GTDB/nr: $94 \pm 4\%/63.6 \pm 16.5\%$ (mean ± standard deviation) on phylum rank, $28.9 \pm 11.2\%/3 \pm 3.2\%$ on genus rank, supplementary Table 5). Considering all data, RAT annotated a larger fraction of reads down to genus rank using GTDB (GTDB/nr: $28 \pm 4.1\%/$

$3.7 \pm 1.9\%$). On species rank, RAT annotated a slightly larger fraction of the reads using nr compared to GTDB (GTDB/nr: $14.7 \pm 2.2\%/16.2 \pm 2.4\%$). This is the result of incomplete taxonomic annotations in nr, where floating species are annotated that have not been assigned to a genus.

In this study, we presented the Read Annotation Tool (RAT), a tool to strengthen the CAT pack metagenome analysis suite. We showed how annotating each read by using the best available taxonomic

information (based on MAGs, contigs, or direct read mapping) leads to fewer falsely detected taxa and improves the accuracy of taxonomic profiles. RAT is flexible to future improvements in sequencing technologies, as well as in assembly and binning software, as they are run by the user before the mapping and classification steps. RAT will be useful in the exploration and understanding of metagenomic datasets by robust classification of most sequencing reads, even in unexplored environments that are rich in novel microorganisms.

## Methods

The biological samples for this research have been collected in accordance with Dutch laws and in cooperation with the company Vitens, which owns the monitoring wells.

### RAT workflow

Read Annotation Tool (RAT) provides individual metagenomic sequencing reads with the most reliable taxonomic annotations and uses these results to reconstruct an accurate taxonomic profile of the microbiome. RAT requires an input of sequencing reads, de novo assembled contigs or scaffolds, and optionally affiliated MAGs. We advise to filter the MAGs based on quality and only supply MAGs that have low contamination (<10%). Completeness of MAGs is less critical, as multiple contigs of the same organism carry a stronger taxonomic signal than individual contigs even if a part of the genome is not binned. These different DNA sequences are queried against a protein database for taxonomic annotations. Next, taxonomic annotations of individual reads are based on the associated data type with the highest confidence of annotation (MAGs > unbinned contigs > unassembled reads). RAT can be run in two different modes: -mcr (MAGs-contigs-reads; complete workflow, see below) and -mc (MAGs-contigs; skips step 3, only evidence from MAGs or contigs is used). The complete workflow of RAT consists of five steps:

1. RAT maps the reads back to the assembled contigs using BWA-MEM[50]. Reads mapping to each contig are extracted with SAMtools[51], including only primary mappings and excluding low-quality primary mappings (default: Phred quality score of 2, which can be changed by the user). In the case of multiple mappings with equal Phred scores, one of the mappings is assigned at random.

2. RAT performs taxonomic annotation of the contigs and MAGs with the previously published tools CAT and BAT[21], respectively. CAT and BAT annotate contigs and MAGs by predicting open reading frames (ORFs) with Prodigal[32] and comparing these with DIAMOND blastp to the non-redundant protein database of NCBI (nr)[36] or the non-redundant set of proteins in GTDB[37], both of which can be downloaded and prepared by running 'CAT download' and 'CAT prepare'. MAGs consist of binned contigs, and therefore, a contig in a MAG gets assigned both a BAT and a CAT annotation that may not be identical. As a MAG contains more taxonomic signals than a contig, RAT will prioritize the MAG annotation. In most metagenomic datasets, not all contigs are binned, and not all contigs can be annotated with CAT[21]. By default, RAT runs CAT with standard settings and BAT with an f parameter value of 0.5. CAT computes a score for each taxonomic assignment it provides, which, when summed, cannot exceed 1 (e.g. if one taxon has a score of 0.7, another taxon can maximally have a score of 0.3 for the same sequence). Using $f < 0.5$ enables it to report multiple annotations per contig/MAG while using $f > 0.5$ prevents multiple annotations per contig/MAG (see ref. 21 for details). Currently, f values < 0.5 are not supported by RAT.

3. Contigs that are not classified and reads that could not be mapped to any contig in step 1 are now classified simultaneously by comparing them to the protein database using DIAMOND blastx[33] and assigning the taxon of the last common ancestor of the organisms found within a certain range of the top hit (default: hits

within 10% of the top-hit bit-score, which can be changed by the user), similar to the r parameter in CAT[21]. Thus, these direct mappings do not involve ORF predictions as in step 2.

4. Each individual read is classified according to the taxonomic signal with the highest confidence, in the following order: (i) If the read is mapped to a contig that is binned, the MAG annotation is assigned to it. (ii) If the read is mapped to an unbinned contig, the contig annotation is assigned to it. (iii) If a read is mapped to an unbinned contig that could not be annotated with CAT or not mapped at all, the direct annotation is assigned to it (see step 3). (iv) Reads that do not have any taxonomic annotation are binned in an 'unclassified' category.

5. RAT calculates the abundance of a taxon by summing the total number of reads assigned to it and normalizes abundances by dividing by the total number of sequenced reads in the sample. This final table constitutes the taxonomic profile. The relative abundances are sequence abundance (fraction of sequenced DNA), as opposed to taxonomic abundance (genome copies)[35]. A user may convert a fraction of sequenced DNA to an estimate of genome copies by normalizing by genome size. The CAT pack provides a table with weighted mean genome sizes for most known bacteria and archaea at all taxonomic ranks based on genomes deposited in the BV-BRC (previously PATRIC) database[48], which allows a user to do this conversion.

RAT is written in Python 3.8.3 and available on GitHub at: https://github.com/MGXlab/CAT_pack. We have tested RAT in the following configuration: BWA v0.7.17, SAMtools v1.10, prodigal v2.6.3, DIAMOND v2.0.5.

### Benchmarking on simulated datasets

To evaluate RAT's performance as read classifier and taxonomic profiler, we used datasets generated for the second Critical Assessment of Metagenome Interpretation (CAMI2) challenge[38]. We used 28 randomly selected samples from three different datasets (mouse gut, marine, rhizosphere), which contain between 78–381 species each. We taxonomically annotated the reads with RAT and four other commonly used profilers: Bracken, Centrifuge, Kraken2, and Kaiju. All tools included in this benchmark also report relative abundance as sequence abundance, and a comparison to RAT is thus fair[35].

For each read, we assessed at six taxonomic ranks (phylum, class, order, family, genus, and species) whether it was correctly or incorrectly annotated or unclassified. To evaluate the taxonomic profiles, we used the same measures used in the CAMI challenge[20]: the L1 and weighted UniFrac distances between the true and inferred profiles on the taxonomic tree, as well as the precision and sensitivity of detected taxa. We only counted taxa as detected if they had been assigned at least 0.001% of the reads in the taxonomic profile and applied the same cut-off for all tools. L1 and weighted UniFrac are pairwise similarity measures between taxonomic profiles. L1 ranges from 0 (profiles are identical) to 2 (profiles do not share any taxa) according to the equation:

$$L1 = \sum_{i=1}^{n} |p1_i - p2_i| \qquad (1)$$

where i is the ith out of n total taxa in the union of the two profiles, and $p1_i$ and $p2_i$ are its relative abundances in the profiles that are being compared[20]. L1 is calculated at each taxonomic rank, contrary to the weighted UniFrac distance. The weighted UniFrac distance incorporates both the relative taxonomic relatedness between taxa and their abundance. We calculated weighted UniFrac distances using EMDUniFrac[52] using the taxonomy as a measure for relatedness with a distance of 1 between taxonomic ranks. Precision and sensitivity are defined as in ref. 20 and only depend on the binary detection of each

organism and not on their abundance. They were calculated using Eqs. (2) and (3):

$$precision = \frac{TP}{TP + FP} \tag{2}$$

$$sensitivity = \frac{TP}{TP + FN} \tag{3}$$

where TP (true positives) is the number of taxa that are correctly detected, FP (false positives) is the number of taxa that are incorrectly detected, and FN (false negatives) is the number of taxa that are not detected but are present in the dataset and thus should have been detected.

We ran Bracken v2.6.1 (ref. [40]), Kraken2 v2.1.2 (ref. [11]), Centrifuge v1.0.4 (ref. [12]), and Kaiju v1.8.2 in greedy mode (ref. [13]) using default settings. A Snakemake implementation of the tool versions and conda environments is available on GitHub at https://github.com/thauptfeld/RAT_paper for reproducibility. We ran all tools using the nr/nt databases from 08th January 2019 that were provided with the CAMI2 challenge.

### Rarefaction curves
Rarefaction curves were calculated for RAT -mcr, RAT -mc, and Kaiju results. We randomly sampled 1%, 5%, 10%, 20%, 30%, 40%, 50%, 60%, 70%, 80%, 90%, and 100% of all reads ten times and counted the number of taxa detected in these subsets.

### Biological datasets
To demonstrate the performance of the RAT workflow on real-world data, we sequenced metagenomes from groundwater, a relatively unexplored biome[53]. 18 samples were collected from three groundwater monitoring wells in an agricultural area in the Netherlands (an overview of the samples can be found in Supplementary Table 3). The samples were collected in accordance with Dutch laws and in collaboration with Vitens, who owns the groundwater monitoring wells. The same samples were used in an earlier study where they were analyzed with 16S rRNA amplicon sequencing[39]. Each well was sampled at six discrete depths between 12 and 64 m below the surface. We filtered 5–7 L of groundwater through 0.2 μm filters (Merck Group, Darmstadt, Germany, catalog number: GSWP14250) and extracted DNA from the filters using the DNeasy PowerSoil Kit (Qiagen, Hilden, Germany, catalog number 47014) according to the manufacturer's instructions. DNA quality (average molecular size) was checked with 1% (w/v) agarose gels stained with 1× SYBR® Safe (Invitrogen, Grand Island, NY, catalog number S33102) and quantified using the dsDNA HS Assay kit for Qubit fluorometer (Invitrogen, catalog number Q32854). Samples were neither diluted nor concentrated before sequencing. Whole metagenome shotgun sequencing was performed on the DNA by Novogene in Hong Kong on the Illumina MiSeq Platform, generating 35,289,790–58,902,006 paired-end sequencing reads of 2 × 251 bp per sample (Supplementary Table 1).

For quality control, assembly, and binning, we used the ATLAS pipeline v2.4 (ref. [54]). ATLAS uses BBTools (https://sourceforge.net/projects/bbmap/) to remove PCR duplicates and adapters and to trim the reads, assemble the reads using SPAdes v3.13.1 in metagenomic mode[24], and bins the contigs using MetaBAT 2 v2.14 (ref. [26]) and MaxBin 2 v2.2.7 (ref. [27]), after which DASTool v.1.1.2 (ref. [44]) is used to optimize MAGs resulting from the two binning approaches. We used MAGs of medium to high quality (>50% completeness, <10% contamination[45]) based on CheckM estimates in '--lineage_wf' mode[43]. We ran RAT on multiple samples at a time using GNU parallel v20210622 (ref. [55]) with the nr database downloaded on the 4th of March 2020., and GTDB release 202. We ran Kaiju on the reads using default settings with a database containing NCBI nr for

bacteria, archaea, viruses, fungi, and microbial eukaryotes from 24 February 2021.

The N50 of the assembled contigs was between 1435 and 3012 nt per sample, the L50 was between 15,196 and 39,823 nt. Out of the 2,770,251 total contigs that were generated from the 18 samples, CAT annotated 2,411,810. All 423 medium- to high-quality MAGs were annotated at superkingdom rank or lower by BAT.

To further assess the diversity of groundwater organisms represented by the MAGs, we dereplicated all medium- to high-quality MAGs with dRep using default settings[46]. We performed a phylogenetic analysis of the dereplicated MAGs based on the CheckM alignment of 43 universal marker genes that are used for phylogenetic placement[43]. A maximum-likelihood phylogenetic tree was inferred with IQ-TREE v2.1.2 (ref. [56]), ModelFinder[57], and UFBoot[58], using the model LG + R10 chosen according to BIC and 1000 UltraFast bootstraps. The resulting tree was visualized with iTOL[59]. The tree was rooted between the archaeal and bacterial MAGs based on their BAT classification (by RAT).

### Plotting
All figures were made using R v4.1.3 and RStudio v1.1.456. The packages used for plotting were ggplot2 (ref. [60]), tidyverse[61], reshape2 (ref. [62]), ggalluvial[63], dplyr (https://dplyr.tidyverse.org), tidyr (https://tidyr.tidyverse.org), RColorBrewer (see http://colorbrewer2.org), Hmisc (https://hbiostat.org/R/Hmisc/), vegan[64], ape[65], and gridExtra (http://CRAN.R-project.org/package=gridExtra).

### Statistics and reproducibility
To pick random samples from the CAMI2 datasets, we used the Python function random.sample(). No statistical method was used to predetermine the sample size. Two rhizosphere samples (samples 8 and 15) were excluded from the study because >1000 taxa of the CAMI2 reference were not present in the database. The experiments were not randomized. The investigators were not blinded to allocation during experiments and outcome assessment.

### Reporting summary
Further information on research design is available in the Nature Portfolio Reporting Summary linked to this article.

## Data availability
All relevant data supporting the key findings of this study are available within the article and its Supplementary Information files. The biological sample data generated in this study have been deposited in the SRA database under BioProject ID PRJNA947390. Data from the CAMI2 challenge is available at https://data.cami-challenge.org/participate. Source data are provided in this paper.

## Code availability
RAT is available on GitHub at https://github.com/MGXlab/CAT_pack (ref. [66]). The scripts used in the downstream analyses are available on GitHub at https://github.com/thauptfeld/RAT_paper and on Zenodo at https://doi.org/10.5281/zenodo.10731871 (ref. [67]). The Snakemake pipeline used to run Centrifuge, Kaiju, and Kraken2/Bracken is available on Zenodo at https://doi.org/10.5281/zenodo.10732074 (ref. [68]).

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

## Acknowledgements

This work was supported by the European Research Council (Consolidator Grant 865694: DiversiPHI to B.E.D.), the Deutsche Forschungsgemeinschaft under Germany's Excellence Strategy (EXC 2051; Project-ID 390713860 to B.E.D.) and the Alexander von Humboldt Foundation in the context of an Alexander von Humboldt Professorship founded by the German Federal Ministry of Education and Research (to B.E.D.). We thank Nora B. Sutton and her group for providing us with groundwater metagenomes to use for this manuscript. We thank Jan Kees van Amerongen for critical technical support. We thank the members of TBB at the University of Utrecht for their valuable input on the text.

## Author contributions

E.H. developed RAT, integrated it into the CAT pack, performed the CAMI2 benchmarks, and wrote the manuscript. N.P. integrated the GTDB option into CAT preparation and wrote some of the downstream analysis code. S.v.I. and B.L.S. performed the analysis on biological data. A.A.-V. sampled the groundwater, extracted DNA, and sent the groundwater samples for sequencing. B.E.D. and F.A.B.v.M. supervised the research and co-wrote the manuscript.

## Funding

## Competing interests

The authors declare no competing interests.
