## [Peer Review File · Nature Communications]

Integrating taxonomic signals from MAGs and contigs improves read annotation and taxonomic profiling of metagenomesReviewer #1 (Remarks to the Author):

The authors describe RAT, a method for quantifying sequence abundance in shotgun metagenomic data building on top of their previously published methods CAT for contig annotation and BAT for MAG annotation. Having observed that the reliability of taxonomic identification increases with sequence length, and thus from read to contig to MAG, the authors propose a tiered approach for assigning reads to taxa: Reads mappable to assembled contigs will be counted towards the first available of 1) a BAT MAG annotation, 2) a CAT contig annotation or 3) a DIAMOND contig annotation. Reads with annotation only directly via DIAMOND will be counted this. The authors proceed to evaluate their implementation against the quantification tools, Bracken, Kaiju, Centrifuge, and Kraken 2, and find it performing favorably.

Major Issues:

RAT in robust mode is, as I understand the manuscript, essentially equivalent to the common assembly based approach where reads are assembled into contigs, contigs binned and classified and reads assigned to taxa via read mapping, which even for a tiered BAT/CAT annotation of contigs is relatively simple to implement. That is not to say that a simple solution is not a good solution, to the contrary. Especially straight forward, seemingly obvious, approaches are often never evaluated rigorously, and solidly crafted, simple methods have great value and longevity in our field. However, this should be appropriately explained in the introduction and the evaluation should be improved by also contrasting RAT-robust with similar, assembly based methods, in addition to the direct-to-reference based methods.

RAT in sensitive mode then straddles the common approaches of assembly based and direct-to-reference, using the latter only as a fall-back, which indeed is novel (to my knowledge). Again, this should be made more explicit in the manuscript introduction and evaluation and the improvement achieved by making use of the otherwise unassigned reads to overall taxonomic profile accuracy should be illustrated better.

I realize that the additional reference methods I am asking for may be simply workflows or reproductions of methods implemented in non-method publications and may be very similar in performance to RAT-robust. However, I believe the manuscript would benefit from improved and more clear positioning of both methods and implementation as well as more weight given to informative evaluation of the methods.

Minor Issues:

Although I will admit that this can be said of any manuscript at any stage, the manuscript could benefit from some additional editing for clarity. I had to re-read several sentences and paragraphs in the results sections a few times to (hopefully) understand them correctly.

Similarly, some of the figures were not as immediately helpful as I would have wished. It may be possible to omit higher taxa, given that the figures only confirm the expectation that misidentifications at these ranks are rare, reducing the number of shapes necessary. Perhaps

some inspiration for figure styles easily interpretable by members of the field can be drawn from the CAMI II paper. (I hesitate to ask, but using CAMI II over CAMI as benchmark would certainly be nice to be fully up to date.)

The table comparing performance between RAT and direct classifiers is, despite the authors' argument that assembly is typically performed anyway, somewhat misleading. Perhaps the total time including assembly can be added in parentheses or as alternate columns. This would become necessary anyway if other, assembly based methods are included in evaluation.

For the purpose of this review I have assumed that the source code matching the manuscript is contained in the "RAT_dev" branch of the referenced git repository. It would have been helpful had the branch already been merged. I would ask for the README.md serving as CAT/BAT/RAT manual to be updated to more clearly explain the different modes.

Reviewer #2 (Remarks to the Author):

The manuscript "Integration of taxonomic signals from MAGs and contigs improves read annotation and taxonomic profiling of metagenomes" addresses the performance enhancement that can be accomplished by breaking data up into fractions with differing quality. This approach is common sense and the authors demonstrate the value of following it. Overall the manuscript is quite well written and addresses the issues involved in such a study quite effectively. I have just a few points to make for which I would like a response.

Major question:

(1) The authors perform the analysis against the NCBI nr sequence database and its taxonomic classification. A modern alternative is also made available with the GTDB as reference sequence database and its taxonomic classification. This is good, but the GTDB reference likely offers different results from the NCBI reference and the performance of the method and how the results change when GTDB is used instead of NCBI reference is not addressed by the study. Both the effects of using a different taxonomic namespace and the sensitivity and precision of the classification against a different reference DB should be analyzed and discussed. It should be as many users will want to use the GTDB reference and taxonomy with the RAT method.

Minor issues:

(2) It wasn't quite clear how the approach worked early enough in the manuscript. For example, on page 4 lines 91-93 should explain more clearly that it is the gene calls in the MAGs and contigs that are used to search against the reference DB, followed by taxonomic assignment smoothing for the contig or MAG and then abundance calculation by read mapping. The details should be left for the Methods section of course, but I found myself guessing about the mechanism of the approach as several options could have been used here, such as whole genome DNA alignment to reference genomes, kmers, etc.

(3) In the analysis "Including taxonomic signals from MAGs and contigs improves read annotation" on page 4 starting at line 110, there should be a statement on the MAG quality threshold used and how it was obtained.

(4) I think, but it wasn't explicitly stated, that the contig-only performance testing is still using contigs that are placed into MAGs, just without applying the taxonomic smoothing from BAT. This should be made clear.

(5) On page 5, line 153, the sentence "Concluding, ..." is not good grammar.

(6) On page 5, lines 163-166, there should be citations for UniFrac, Centrifuge, Kaiju, Kraken2, and Bracken.

(7) On page 5, line 168, I would like to know how many reads typically correspond to a relative abundance of 0.001%. Does that really just mean 1 read by itself isn't considered valid? What about if there is taxonomic smoothing to higher levels in the read-based taxonomic annotation? One could imagine discarding strays that collectively exceed this threshold when grouped and assigned at a higher taxonomic level as would happen for more novel lineages and even for hits to closer neighbors in the reference DB. I suspect the thresholding is done after that grouping but this should be made clear in the text.

(8) Page 6 line 183 and page 7 line 185. I'm a bit confused why Kaiju is performing as well as RAT. A read-only approach to annotation absolutely should not be as good as a MAG and contig approach where full-length genes are used for homology detection. I worry that the Kaiju database has genomes that closely match to the lineages in the read test data and it makes me suspect the entire study. Please address how the reference databases were cleaned of all references too close to the test data to ensure a fair analysis.

(9) Page 7 lines 200-209. I wanted a bit more clarity on how reads are mapped, how taxonomic smoothing is done, and how higher ranks are assigned. Not full details, just a few more words and then a mention that the details are in the Methods section.

(10) Page 7, line 203. Please commit to a sentence. I vote for "... the surrounding regions of the genome are considered ..." :-)

(11) Page 7, line 214. The fraction of the test data that falls within MAGs is very unusual. I don't want to add a lot more work but I think a fair picture of the performance improvement your approach offers on normal samples with a lower proportion of highly abundant lineages is justified. Perhaps the CAMI datasets don't individually offer this but what if you add the unassembled reads from all the CAMI datasets on top to reduce the fraction contributed by the abundant lineages winding up in contigs and MAGs?

(12) Page 7, line 227. Please state that the contigs and MAGs are from CAMI-6 and CAMI-13. It's implicit, but make it explicit.

(13) Page 10, lines 308-310. The point that these are not lineage abundances but read abundances that can be corrected with genome size should be made earlier in the manuscript.

(14) Page 11, lines 365 and Page 12 line 393. A statement that default parameters were used is insufficient. I know this is somewhat common practice but I'm against it. Please give the command line with all parameters for each method.

Reviewer #3 (Remarks to the Author):

== Introduction

The authors have developed a novel method for the taxonomic profiling of microbiomes using various metagenomic data. Their software comes with certain modes (sensitive, robust) to address the problem of data "overexplanation", which can lead to wrong or at least unsafe taxonomic assignments. They rigorously benchmark their new method against proven benchmarking dataset as well as compare it to other established methods in the field. For example, according to their data, the precision of the new method(s) clearly outperforms those of other established methods and is in line with earlier findings of the group. The presented results indicate that the authors' new method could indeed fill a known gap in the field of taxonomic profiling.

== Potential critique

Based on the presented data, I have not found any severe problems, rather than some conceptual issues that might require further discussion (see below) and/or clarification. Due to memory limitations on my machine (32GB RAM machine) and a lack of suitable server infrastructure at the moment, it wasn't possible for me to test-run RAT (i.e. the current version on Github), even though RAT robust has one of lowest memory footprints among the tested tools, which is very good.

== Major issues (i.e. need for clarification)

* I. 121: Correctly assigning sequences to a taxonomic rank not only requires a robust method but also an existing taxonomic classification against one can annotate the samples to (i.e. correct names). If the classification is e.g. outdated or not properly curated the taxonomic assignments at the different ranks will be negatively affected by that. Now, since this part was not entirely clear to me while reading the manuscript, could the authors please discuss this aspect e.g. in view of authoritative classifications as e.g. provided by LPSN? From the technical documentation found at <https://github.com/MGXlab/CAT> I figure that CAT/BAT/RAT is using the NCBI taxonomy and that is not an authoritative one (e.g. see the disclaimer at <https://www.ncbi.nlm.nih.gov/Taxonomy/Browser/wwwtax.cgi>). I understand that at some point a mapping of protein sequences to taxon names has to be provided and that the NCBI data dumps are providing this info in a convenient way but still the aforementioned names can be wrong. This should be discussed/clarified.

* I. 383: When UniFrac incorporates the relative phylogenetic relatedness, how does it deal with unresolved trees (i.e. unsupported background trees)? I browsed through some implementations of UniFrac (<https://www.nature.com/articles/ismej200997>), including EMDUniFrac (<https://doi.org/10.1007/s00285-018-1235-9>), but couldn't find information on that matter. Now, should such important information not be considered by these approaches, this has the potential to heavily skew the explanatory power of this type of distance metrics. Of course, all results compared via UniFrac would be affected by that but I think the authors should critically discuss this problem.

== Minor issues

* I. 26: I recommend to introduce the long form of CAT and BAT as well.

* I. 26: On the Github page there is no mention of RAT. Is it already integrated in the CAT/BAT workflow (changelog says no)?

* I. 31: To the best of my knowledge this is not quite correct. HGT was found to be present in specific, oftentimes closely related organismal groups. But it is certainly not a general issue (<https://www.pnas.org/doi/full/10.1073/pnas.1632870100>). Barriers to HGT not only exist at higher taxonomic ranks (as correctly stated by the authors in I. 211) (<https://academic.oup.com/gbe/article/15/6/evad089/7180211>).

* I. 39: In the main text, the term "annotation" is mentioned here for the first time and it might be confusing for those who are more used to terms like "classification of organisms based on their sequence data" instead of "annotation of sequences". It might be a good idea to precisely define once what the authors exactly mean by the term "annotation".

* I. 59: write out "(data explained: reads > contigs > MAGs)"

* I. 67: write out "(MAGs > contigs > reads)"

* I. 90: here or elsewhere: please state which one of DIAMOND's built-in sensitivity modes you chose while running it in blastp or blastx mode. The choice of mode likely has a huge impact on the subsequent results. According to the technical documentation the use of DIAMOND's "--sensitive" parameter is possible.

* I. 111: In addition to the previous point: First, the benchmarking will only be reproducible if all parts of the pipeline are properly described in terms of the parameter settings used. Descriptions such as "the program x was run using default parameters" might not be sufficient as default parameters might vary between versions. Second, users are supposed to use the new tool under the settings which were shown in this study to outperform the other tools (How to safeguard

this?). If other settings are used instead, the results of the tool might not be optimal anymore.

- * l. 142: comment: this conservative decision is reasonable
- * l. 163: "CAMI ground truth taxonomic profiles", maybe better: "CAMI taxonomic reference profiles"
- * l. 245: It is probably better to use "unrecognizables" instead of "unknowns"? Later you could replace "unknownness" by "unrecognizability?"
- * l. 263: What type of phylogenetic tree was inferred? Which inference method was used etc.? Number of replicates? Info can also be added to the Supplementary Figure 5.
- * l. 272: check proper italicization of taxon names throughout the manuscript
- * l. 343: The "f parameter" does what? Please provide a brief description for those who are not familiar with this term.
- * l. 410: It would be helpful to write out that an overview of all samples is found in Supplementary Table 1.

== Figures:

- * e.g. in Fig. 2 the yellow and orange colors are difficult to distinguish
- * median mark on the bar charts only barely visible for "RAT robust"

== Tables:

- * l. 226: What about disk space usage during the various runs? The default database already uses c. 180 GB.
- * l. 227: The prefix of the simulated datasets "smp" stands for what?
- * l. 228: "bracken" -> "Bracken"
- * l. 229: 16 parallel CPUs or 16 CPU cores? Please mention the CPU types and number of cores.

== Supplementary Data

- * Generally I recommend to not use a MS Word file but to rather render a proper PDF (e.g. using LaTeX) in which the images are included as vector graphics to allow the user to zoom in the more complex figures without loss of quality. For example, the text in Suppl. Fig. 5 is not readable for me.
- * l. 20: What do the authors mean by "[...] and the taxonomic signal they originate from"? Do they perhaps mean "water sample" instead?
- * l. 24 (Supplementary Fig. 5): What is the meaning of the background colors behind the leaf labels?
- * l. 32 (Supplementary Fig. 6): "CAMI truth" sounds a bit odd. Maybe better to use "CAMI reference"

== Language issues (typos, grammar etc.)

The language is very good.

Rebuttal Letter NCOMMS-23-36809-T

REVIEWER COMMENTS

Reviewer #1 (Remarks to the Author):

1.01 - The authors describe RAT, a method for quantifying sequence abundance in shotgun metagenomic data building on top of their previously published methods CAT for contig annotation and BAT for MAG annotation. Having observed that the reliability of taxonomic identification increases with sequence length, and thus from read to contig to MAG, the authors propose a tiered approach for assigning reads to taxa: Reads mappable to assembled contigs will be counted towards the first available of 1) a BAT MAG annotation, 2) a CAT contig annotation or 3) a DIAMOND contig annotation. Reads with annotation only directly via DIAMOND will be counted this. The authors proceed to evaluate their implementation against the quantification tools, Bracken, Kaiju, Centrifuge, and Kraken 2, and find it performing favorably.

We thank Reviewer #1 for their time and effort reviewing our study and their comments. We appreciate the constructive suggestions and hope that we have addressed all their concerns accordingly.

Major Issues

1.02 - RAT in robust mode is, as I understand the manuscript, essentially equivalent to the common assembly based approach where reads are assembled into contigs, contigs binned and classified and reads assigned to taxa via read mapping, which even for a tiered BAT/CAT annotation of contigs is relatively simple to implement. That is not to say that a simple solution is not a good solution, to the contrary. Especially straight forward, seemingly obvious, approaches are often never evaluated rigorously, and solidly crafted, simple methods have great value and longevity in our field. However, this should be appropriately explained in the introduction and the evaluation should be improved by also contrasting RAT-robust with similar, assembly based methods, in addition to the direct-to-reference based methods.

We agree that our approach is straightforward. However, as mentioned below by reviewer #1, the approach is usually contained in a collection of in-house scripts that can only be run by people with good understanding of the main programming language(s) used in the lab. One of our goals was to make best practice assembly-based profiling of shotgun metagenomes more accessible to non-computational biologists and then building into it the use of direct read annotations for unmapped reads.

We have made the fact that assembly-based profiling is an established method more explicit by adding the following text to the introduction (from line 57): "Thus, assembly-based annotation of MAGs and contigs comprise a current best practice for analysing shotgun metagenomic datasets. The reliable taxonomic annotations together with read-based coverage of the MAGs and contigs can be used to estimate taxonomic profiles that have little noise and high explanatory power." We cite studies that benchmarked the value of assembly-based profiling methods and a study that used this method to profile the metagenome of a groundwater treatment pipeline.

1.03 - RAT in sensitive mode then straddles the common approaches of assembly based and direct-to-reference, using the latter only as a fall-back, which indeed is novel (to my knowledge). Again, this should be made more

explicit in the manuscript introduction and evaluation and the improvement achieved by making use of the otherwise unassigned reads to overall taxonomic profile accuracy should be illustrated better.

We have added/adapted the following text to the manuscript to address this comment:

- From line 74: “Direct read annotation by DIAMOND is then used for those reads that cannot be associated with a contig or MAG and to improve the sensitivity of the resulting taxonomic profile.”
- From line 215: “RAT -mcr slightly outperformed RAT -mc in L1 distance and sensitivity, indicating that including direct read annotation leads to reconstructed profiles that are more similar to the reference profile than when relying solely on assembly-based profiling.”
- From line 264: “Including direct read annotation consistently increased RAT’s sensitivity compared to RAT -mc on all ranks and across all three datasets. One may expect that these classifications are less robust than those annotated via MAGs or contigs.”

1.04 - I realize that the additional reference methods I am asking for may be simply workflows or reproductions of methods implemented in non-method publications and may be very similar in performance to RAT-robust. However, I believe the manuscript would benefit from improved and more clear positioning of both methods and implementation as well as more weight given to informative evaluation of the methods.

We agree with the reviewer and would like to refer to our responses to comments 1.02 and 1.03.

Minor Issues

1.05 - Although I will admit that this can be said of any manuscript at any stage, the manuscript could benefit from some additional editing for clarity. I had to re-read several sentences and paragraphs in the results sections a few times to (hopefully) understand them correctly.

We have rechecked the manuscript thoroughly ourselves and with the help of colleagues and rephrased/shortened/split any sentences that might be confusing. Please see the document with tracked changes.

1.06 - Similarly, some of the figures were not as immediately helpful as I would have wished. It may be possible to omit higher taxa, given that the figures only confirm the expectation that misidentifications at these ranks are rare, reducing the number of shapes necessary. Perhaps some inspiration for figure styles easily interpretable by members of the field can be drawn from the CAMI II paper. (I hesitate to ask, but using CAMI II over CAMI as benchmark would certainly be nice to be fully up to date.)

We agree with reviewer #2 that the CAMI2 datasets should be included in the benchmark. The dataset used for the original benchmark was indeed from CAMI2, albeit one of the toy datasets (mouse gut). We have made this more explicit, by changing any mentions of CAMI to CAMI2 where appropriate and adding the phrase “the second round of the CAMI challenge” in other places in the manuscript. At the time when we ran the initial benchmark, the official references for the CAMI2 challenge datasets were not yet available. We have now added two of the CAMI2 real-world datasets to our benchmark - marine and rhizosphere. We discuss them in the sections “Including taxonomic signals from MAGs and contigs improves read annotation” and “Including information from contigs

and MAGs improves accuracy of taxonomic profiling” and show the results in Supplementary Figures 2-5. A table containing important information about the datasets is included as Supplementary Table 1.

We have also changed the original Figure 2 to make the information contained more accessible, and removed the class, order, and family ranks for the figures (the figures containing all ranks is still available as Supplementary Figure2).

1.07 - The table comparing performance between RAT and direct classifiers is, despite the authors' argument that assembly is typically performed anyway, somewhat misleading. Perhaps the total time including assembly can be added in parentheses or as alternate columns. This would become necessary anyway if other, assembly-based methods are included in evaluation.

We see how the argument about the assembly being performed anyway could be misleading. We have added the time it took to assemble and bin the two samples in Supplementary Table 2, which we also refer to in the text.

1.08 - For the purpose of this review I have assumed that the source code matching the manuscript is contained in the "RAT_dev" branch of the referenced git repository. It would have been helpful had the branch already been merged. I would ask for the README.md serving as CAT/BAT/RAT manual to be updated to more clearly explain the different modes.

We have since merged the branches on Github and updated the README.md to include instructions for running RAT including code examples.

Reviewer #2 (Remarks to the Author):

2.01 - The manuscript "Integration of taxonomic signals from MAGs and contigs improves read annotation and taxonomic profiling of metagenomes" addresses the performance enhancement that can be accomplished by breaking data up into fractions with differing quality. This approach is common sense and the authors demonstrate the value of following it. Overall the manuscript is quite well written and addresses the issues involved in such a study quite effectively. I have just a few points to make for which I would like a response.

We thank reviewer #2 for the compliments on our manuscript and appreciate the valuable insights we have gained from their comments. We hope that our responses address all the points raised by reviewer #2 appropriately.

Major question

2.02 - The authors perform the analysis against the NCBI nr sequence database and its taxonomic classification. A modern alternative is also made available with the GTDB as reference sequence database and its taxonomic classification. This is good, but the GTDB reference likely offers different results from the NCBI reference and the performance of the method and how the results change when GTDB is used instead of NCBI reference is not addressed by the study. Both the effects of using a different taxonomic namespace and the

sensitivity and precision of the classification against a different reference DB should be analyzed and discussed. It should be as many users will want to use the GTDB reference and taxonomy with the RAT method.

In the manuscript, we have now included a comparison of using RAT with NCBI vs. GTDB taxonomy on the biological datasets in a new section called “GTDB compatibility provides lower rank annotations on biological data”. We decided against including the comparison in the section about the CAMI2 benchmark, because the CAMI2 “ground truth” was based on NCBI Taxonomy. Thus, a “true” annotation with GTDB is not available for the CAMI2 data and thus comparing the sensitivity/precision of the annotations would be misleading. We did include a comparison of the number of annotated reads/contigs/MAGs in the CAMI2 datasets in the new Supplementary Figure 11 and Supplementary table 5.

Minor issues

2.03 - It wasn't quite clear how the approach worked early enough in the manuscript. For example, on page 4 lines 91-93 should explain more clearly that it is the gene calls in the MAGs and contigs that are used to search against the reference DB, followed by taxonomic assignment smoothing for the contig or MAG and then abundance calculation by read mapping. The details should be left for the Methods section of course, but I found myself guessing about the mechanism of the approach as several options could have been used here, such as whole genome DNA alignment to reference genomes, kmers, etc.

We have adapted the paragraph that explains the algorithm and changed the following sentences (starting in line 97): “CAT and BAT predict ORFs on these longer sequences and query them against a protein reference database with DIAMOND blastp. Taxonomy of the sequence is assigned based on the combined taxonomic signal of the individual ORFs, selecting higher-ranking taxa in cases where many conflicting signals are present. Default options for the reference database include the NCBI non-redundant protein database (nr) and, in the latest RAT update, the non-redundant set of proteins in the Genome Taxonomy Database (GTDB). Alternatively, any protein database with taxonomic annotations can be supplied by the user. Next, individual reads are mapped to the contigs with BWA-MEM, and each read inherits the taxonomic annotation with the highest reliability: the MAG annotation if the contig is binned, and the contig annotation if it is unbinned. Finally, the remaining sequences (reads that do not map to a contig and contigs that cannot be annotated by CAT) are annotated individually by querying them directly against the protein database with DIAMOND blastx in default sensitivity mode.”

2.04 - In the analysis “Including taxonomic signals from MAGs and contigs improves read annotation” on page 4 starting at line 110, there should be a statement on the MAG quality threshold used and how it was obtained.

We have added the statement “(<10% contamination)” in the sentence starting from line 144.

2.05 - I think, but it wasn't explicitly stated, that the contig-only performance testing is still using contigs that are placed into MAGs, just without applying the taxonomic smoothing from BAT. This should be made clear.

We have changed the sentence describing the RAT runs to: “We compared five different methods for read annotation: (i) RAT -r to annotate all reads directly with DIAMOND blastx, without mapping

them to contigs or MAGs, (ii) RAT -cr for taxonomic annotations via contigs but ignoring MAG annotations, and direct read annotations for reads that did not map to contigs, (iii) RAT -mcr for annotations via MAGs, contigs, and reads, using the MAGs included in CAMI2 ('CAMI genomes'), (iv) RAT -mcr for annotations via MAGs, contigs, and reads, using MAGs binned by MetaBAT2 (<10% contamination), and (v) RAT -mc for annotations via MAGs and contigs, using and MetaBAT2 MAGs, but no direct read annotations."

2.06 - On page 5, line 153, the sentence "Concluding, ..." is not good grammar.

We have changed the beginning of the sentence (now line 184) to "In conclusion,".

(6) On page 5, lines 163-166, there should be citations for UniFrac, Centrifuge, Kaiju, Kraken2, and Bracken.

We have included the citation in the text (currently starting from line 196).

2.07 - On page 5, line 168, I would like to know how many reads typically correspond to a relative abundance of 0.001%. Does that really just mean 1 read by itself isn't considered valid? What about if there is taxonomic smoothing to higher levels in the read-based taxonomic annotation? One could imagine discarding strays that collectively exceed this threshold when grouped and assigned at a higher taxonomic level as would happen for more novel lineages and even for hits to closer neighbors in the reference DB. I suspect the thresholding is done after that grouping but this should be made clear in the text.

There are a few points we would like to mention in response to this comment:

- Reviewer #2 is correct in their assumption that the threshold is applied after taxonomic smoothing. If a species is present below the threshold but its genus has a higher abundance, then the reads belonging to the species do not get included in the taxonomic profile on species level, but they do get included at genus rank and above. We have made this explicit in the manuscript in line 201 by saying "This cut-off was applied separately on every taxonomic rank. Thus, even if a species with an abundance below 0.001% was removed, its reads could still be included at the genus rank if the genus was >0.001%."
- An abundance of >0.001% corresponds to 4 reads in CAMI2 all datasets. We have included this information in the figure legends of figures 3, and supplementary figures 4 and 5, and in the text.
- We believe that the user should get full decision power in the usage of our tool. Therefore, RAT itself does not exclude any taxa, even if they are only represented by one read. The threshold of 0.001% is a recommendation that we make in the manuscript due to the performance that we have seen in the benchmark (and we have applied the same cut-off for the other tools as well, as mentioned in line 201). We have made this more explicit in lines 198 saying, "The RAT output includes all taxa that are represented by at least a single read. However, as direct read annotations are known to be inaccurate, we limited the amount of noise by only considering taxa with a minimum abundance of 0.001% (which represents 4 reads in the CAMI2 samples) and applied this cut-off for all tools in the benchmark."

2.08 - Page 6 line 183 and page 7 line 185. I'm a bit confused why Kaiju is performing as well as RAT. A read-

only approach to annotation absolutely should not be as good as a MAG and contig approach where full-length genes are used for homology detection. I worry that the Kaiju database has genomes that closely match to the lineages in the read test data and it makes me suspect the entire study. Please address how the reference databases were cleaned of all references too close to the test data to ensure a fair analysis.

We agree that the great performance of kaiju on read annotation and taxonomic profiling is striking and understand Reviewer #2's concerns. The dataset originally used in the benchmark (toy mouse gut from the CAMI2 challenge) did in fact only contain taxa with a species representative in the reference database albeit with a high simulated error rate that represents sequence divergence. To improve our benchmark, we added two more CAMI2 datasets to the manuscript, marine and rhizosphere. We write about them in the sections "Including taxonomic signals from MAGs and contigs improves read annotation" and "Including information from contigs and MAGs improves accuracy of taxonomic profiling" and show the results in Supplementary Figures 2-5. A table containing important information about the datasets is included as Supplementary Table 1.

The new datasets contain on average ~15-29% reads from organisms that do not have a species and/or genus representative in the database used (see Supplementary Table 1, Supplementary Figure 2). These datasets confirm the great performance of kaiju in read annotation. Although the True positive rate drops for the reads that do not have a species representative in the database, L1 distance and weighted Unifrac show how well kaiju does as a profiler (Supplementary Figures 4 and 5). However, the marine and rhizosphere datasets also confirm the high rate of false positive taxa in the taxonomic profile by Kaiju as also observed in the real biological groundwater dataset, even on high taxonomic ranks (Supplementary Figures 4 and 5).

2.09 - Page 7 lines 200-209. I wanted a bit more clarity on how reads are mapped, how taxonomic smoothing is done, and how higher ranks are assigned. Not full details, just a few more words and then a mention that the details are in the Methods section.

We now clarify this by adding "with BWA-MEM" to the sentences starting on lines 90, 103. We have also adapted the sentence starting on line 254 to "Reads belonging to novel taxa within known clades are also more likely to get correctly annotated, as when the reads are assembled into contigs or MAGs, RAT may annotate them on a higher taxonomic rank. For example, if the closest related sequences in the reference database are found among different species in a genus, the sequence will be annotated at the genus rank (see Methods)."

2.10 - Page 7, line 203. Please commit to a sentence. I vote for "... the surrounding regions of the genome are considered ..." :-)

Thanks for noting this, we have committed to the sentence as suggested.

2.11 - Page 7, line 214. The fraction of the test data that falls within MAGs is very unusual. I don't want to add a lot more work but I think a fair picture of the performance improvement your approach offers on normal samples with a lower proportion of highly abundant lineages is justified. Perhaps the CAMI datasets don't individually offer this but what if you add the unassembled reads from all the CAMI datasets on top to reduce the fraction contributed by the abundant lineages winding up in contigs and MAGs?

We agree that the fraction of reads ending up in bins in the simulated data is high, and we explicitly mention this in the text. We now also refer to the section on real metagenomes. In that section, we show what realistic numbers look like, and again explicitly make the comparison with the simulated datasets.

2.12 - Page 7, line 227. Please state that the contigs and MAGs are from CAMI-6 and CAMI-13. It's implicit, but make it explicit.

We have adapted the text to "CAMI2 mouse gut samples 6: 33,098,456 reads and 13: 33,184,772 reads".

2.13 - Page10, lines 308-310. The point that these are not lineage abundances but read abundances that can be corrected with genome size should be made earlier in the manuscript.

We have included this fact at the beginning of the Results section in the sentence starting from line 82.

Reviewer #3 (Remarks to the Author):

Introduction

3.01 - The authors have developed a novel method for the taxonomic profiling of microbiomes using various metagenomic data. Their software comes with certain modes (sensitive, robust) to address the problem of data "overexplanation", which can lead to wrong or at least unsafe taxonomic assignments. They rigorously benchmark their new method against proven benchmarking dataset as well as compare it to other established methods in the field. For example, according to their data, the precision of the new method(s) clearly outperforms those of other established methods and is in line with earlier findings of the group. The presented results indicate that the authors' new method could indeed fill a known gap in the field of taxonomic profiling.

We are grateful to reviewer #3 for the time and dedication invested in reviewing our manuscript and we trust that we have responded to all the comments in a suitable manner.

Potential critique

3.02 - Based on the presented data, I have not found any severe problems, rather than some conceptual issues that might require further discussion (see below) and/or clarification. Due to memory limitations on my machine (32GB RAM machine) and a lack of suitable server infrastructure at the moment, it wasn't possible for me to test-run RAT (i.e. the current version on Github), even though RAT robust has one of lowest memory footprints among the tested tools, which is very good.

We understand the constraints the reviewer faced due to the limited memory capacity. Unfortunately, due to the size of the nr and GTDB databases, running RAT is limited to machines with high available RAM. We thank the reviewer for their acknowledgment of RAT's efficiency in this context.

Major issues (i.e. need for clarification)

*3.03 - * I. 121: Correctly assigning sequences to a taxonomic rank not only requires a robust method but also an existing taxonomic classification against one can annotate the samples to (i.e. correct names). If the classification is e.g. outdated or not properly curated the taxonomic assignments at the different ranks will be negatively affected by that. Now, since this part was not entirely clear to me while reading the manuscript, could the authors please discuss this aspect e.g. in view of authoritative classifications as e.g. provided by LPSN? From the technical documentation found at <https://github.com/MGXlab/CAT> I figure that CAT/BAT/RAT is using the NCBI taxonomy and that is not an authoritative one (e.g. see the disclaimer at <https://www.ncbi.nlm.nih.gov/Taxonomy/Browser/wwwtax.cgi>). I understand that at some point a mapping of protein sequences to taxon names has to be provided and that the NCBI data dumps are providing this info in a convenient way but still the aforementioned names can be wrong. This should be discussed/clarified.*

While we agree with the Reviewer that NCBI is not a taxonomic authority like LPSN, it does strive to represent the latest taxonomy and keep it updated. Moreover, it links to a swath of Major 3rd Party Taxonomic Data Resources including LPSN (see <https://www.ncbi.nlm.nih.gov/Taxonomy/taxonomyhome.html/index.cgi?chapter=resources>). Importantly, it provides for each prokaryotic taxon a unique, unambiguous, and stable identifier, which is essential when creating a usable bioinformatic tool. In contrast, using established taxonomic names would immediately lead to problems, e.g. when encountering the infamous genus *Bacteria* (stick insects) in a sample (see <https://www.ncbi.nlm.nih.gov/Taxonomy/Browser/wwwtax.cgi?mode=Undef&id=629395>). Together with GTDB which we have now also included, NCBI Taxonomy is by far the most commonly used resource in our field, and we are confident that it is all a user of our tool will be looking for. To respond to this comment, we have now mentioned the above points at the point where we introduce the taxonomy options provided by RAT (see our response 2.02 to Reviewer #2 above).

*3.04 - * I. 383: When UniFrac incorporates the relative phylogenetic relatedness, how does it deal with unresolved trees (i.e. unsupported background trees)? I browsed through some implementations of UniFrac (<https://www.nature.com/articles/ismej200997>), including EMDUniFrac (<https://doi.org/10.1007/s00285-018-1235-9>), but couldn't find information on that matter. Now, should such important information not be considered by these approaches, this has the potential to heavily skew the explanatory power of this type of distance metrics. Of course, all results compared via UniFrac would be affected by that but I think the authors should critically discuss this problem.*

In the context of our study, the Unifrac distance is calculated over the taxonomic tree, not a phylogenetic tree. We now clarify this in the Methods. Since the taxonomic tree is based on the complete taxonomy provided by CAMI2, all trees are by definition resolved.

Minor issues

*3.05 - * I. 26: I recommend to introduce the long form of CAT and BAT as well.*

We have changed the sentence in line xxx to “The package of the Contig Annotation Tool (CAT), Bin Annotation Tool (BAT), and RAT is available at <https://github.com/MGXlab/CAT>.”

3.06 - * l. 26: On the Github page there is no mention of RAT. Is it already integrated in the CAT/BAT workflow (changelog says no)?

At the time of submission, RAT had been available in its own branch "RAT_dev" on the Github. In the meantime, the branches have been merged and RAT is part of the new CAT release (v.6.0)

3.07 - * l. 31: To the best of my knowledge this is not quite correct. HGT was found to be present in specific, oftentimes closely related organismal groups. But it is certainly not a general issue (<https://www.pnas.org/doi/full/10.1073/pnas.1632870100>). Barriers to HGT not only exist at higher taxonomic ranks (as correctly stated by the authors in l. 211) (<https://academic.oup.com/qbe/article/15/6/evad089/7180211>).

We have edited the sentence to "microbes have high rates of horizontal gene transfer, so the best hit in the reference database might be from a different taxon" in line 42.

3.08 - * l. 39: In the main text, the term "annotation" is mentioned here for the first time and it might be confusing for those who are more used to terms like "classification of organisms based on their sequence data" instead of "annotation of sequences". It might be a good idea to precisely define once what the authors exactly mean by the term "annotation".

We have added the explanation "(i.e., assignment of sequences to a certain taxon)" in line 39.

3.09 - * l. 59: write out "(data explained: reads > contigs > MAGs)"

We have changed the sentence in line 55 to "Long sequence length mitigates the errors in annotation discussed earlier because multiple taxonomic signals can be integrated, as the confidence in the taxonomic annotation is highest in MAGs, followed by contigs and then reads."

3.10 - * l. 67: write out "(MAGs > contigs > reads)"

We have changed the sentence in line 60 to "However, even though taxonomic annotation is more accurate for longer sequences, they often represent only part of the metagenomic data and therefore provide an incomplete picture of the microbiome, since most data is explained by reads, followed by contigs and then MAGs."

3.11 - * l. 90: here or elsewhere: please state which one of DIAMOND's built-in sensitivity modes you chose while running it in blastp or blastx mode. The choice of mode likely has a huge impact on the subsequent results. According to the technical documentation the use of DIAMOND's "--sensitive" parameter is possible.

We have adapted the sentence starting in line 105 to: "Finally, the remaining sequences (reads that do not map to a contig and contigs that cannot be annotated by CAT) are annotated individually by querying them directly against the protein database with DIAMOND blastx in default sensitivity mode."

3.12 - * l. 111: In addition to the previous point: First, the benchmarking will only be reproducible if all parts of the pipeline are properly described in terms of the parameter settings used. Descriptions such as "the program x was run using default parameters" might not be sufficient as default parameters might vary between versions. Second, users are supposed to use the new tool under the settings which were shown in this study to

outperform the other tools (How to safeguard this?). If other settings are used instead, the results of the tool might not be optimal anymore.

To maximize the reproducibility of our study, we have made everything available on GitHub at https://github.com/thauptfeld/RAT_paper: a full snakemake implementation of the entire workflow, including tools in the right versions and with all parameter settings, as well as conda environments. This is now mentioned in the manuscript in the sentence starting at line 479.

*3.13 - * l. 142: comment: this conservative decision is reasonable*

This choice was made in line with the general, conservative strategy of the tools in the CAT/BAT/RAT pack, as we showed that its robust annotations vastly improve the specificity of the tools (see Figure 3 and the CAT/BAT paper).

*3.14 - * l. 163: "CAMI ground truth taxonomic profiles", maybe better: "CAMI taxonomic reference profiles"*

We have changed "CAMI ground truth" to "CAMI2 reference" throughout the manuscript.

*3.15 - * l. 245: It is probably better to use "unrecognizables" instead of "unknowns"? Later you could replace "unknownness" by "unrecognizability?"*

While we acknowledge that terms such as "unknowns" and "unknownness" could be ambiguous to some readers, we opt to keep this terminology, as it is consistent with previous publications, including the CAT/BAT paper (reference 21).

*3.16 - * l. 263: What type of phylogenetic tree was inferred? Which inference method was used etc.? Number of replicates? Info can also be added to the Supplementary Figure 5.*

We have added the number of bootstraps to the sentence starting in line 511: "A maximum-likelihood phylogenetic tree was inferred with IQ-TREE v2.1.2, ModelFinder, and UFBoot, using the model LG+R10 chosen according to BIC and 1000 bootstraps."

*3.17 - * l. 272: check proper italicization of taxon names throughout the manuscript*

We have now italicized all taxon names in the manuscript.

*3.18 - * l. 343: The "f parameter" does what? Please provide a brief description for those who are not familiar with this term.*

We have adapted the sentences starting from line 425 to: "By default, RAT runs CAT with standard settings, and BAT with an f parameter value of 0.5. CAT computes a score for each taxonomic assignment it provides, which, when summed, cannot exceed 1 (e.g. if one taxon has a score of 0.7, another taxon can maximally have a score of 0.3 for the same sequence). Using $f < 0.5$ enables it to report multiple annotations per contig/MAG, while using $f > 0.5$ prevents multiple annotations per contig/MAG (see CAT REF for details). Currently, f values < 0.5 are not supported by RAT."

*3.19 - * l. 410: It would be helpful to write out that an overview of all samples is found in Supplementary Table 1.*

We have added the text “(an overview of the samples can be found in Supplementary Table 3)” to the sentence starting in line 488.

Figures:

3.20 - * e.g. in Fig. 2 the yellow and orange colors are difficult to distinguish

3.21 - * median mark on the bar charts only barely visible for "RAT robust"

We have used a brighter shade of yellow for “RAT -mcr” in Fig. 2.

Tables:

3.22 - * l. 226: What about disk space usage during the various runs? The default database already uses c. 180 GB.

We have added the size of the combined output files to Table 1.

3.23 - * l. 227: The prefix of the simulated datasets "smp" stands for what?

We have changed all instances of “smp” to “sample” in the manuscript.

3.24 - * l. 228: "bracken" -> "Bracken"

We have changed “bracken” to “Bracken”.

3.25 - * l. 229: 16 parallel CPUs or 16 CPU cores? Please mention the CPU types and number of cores.

We have adapted the sentence starting from line 288 to: “All runs were performed using 16 CPU cores when the program allowed for multithreading (Intel Xeon Gold 6240R).”

== Supplementary Data

3.26 - * Generally I recommend to not use a MS Word file but to rather render a proper PDF (e.g. using LaTeX) in which the images are included as vector graphics to allow the user to zoom in the more complex figures without loss of quality. For example, the text in Suppl. Fig. 5 is not readable for me.

We agree that reading the figures inside the Word file is difficult. We have attached high-resolution and/or vector graph versions of all figures to the resubmission.

3.27 - * l. 20: What do the authors mean by "[...] and the taxonomic signal they originate from"? Do they perhaps mean "water sample" instead?

We have added the text “(i.e. MAG, contig, direct read mapping)” to the Figure legend.

3.28 - * l. 24 (Supplementary Fig. 5): What is the meaning of the background colors behind the leaf labels?

We have added the following sentence to the figure legend of Supplementary Figure 9: “Colors behind the leaf labels highlight the members of large clusters of microbes (red - *Archaea*, yellow - *Chloroflexi*, blue - *Candidatus Omnitrophica*, green - *Proteobacteria*).”

3.29 - * l. 32 (Supplementary Fig. 6): "CAMI truth" sounds a bit odd. Maybe better to use "CAMI reference"

We have changed “CAMI truth” and “CAMI ground truth” to “CAMI reference” throughout the manuscript.

=> Language issues (typos, grammar etc.)

3.30 - *The language is very good.*

Thank you.

Reviewer #1 (Remarks to the Author):

My thanks to the authors for addressing my concerns in full!

I have a few suggestions regarding the code distribution (see the new code section below). However, these are to be considered only as suggestions. I believe the manuscript is good for publication as is.

Reviewer #1 (Remarks on code availability):

CAT is implemented in Python as a collection of scripts with each pipeline step implemented in a single script, one file for shared code and an entry point script "CAT" used to dispatch commands. It is valid to do this, staying deliberately simple.

The choice of name for the entry point, "CAT", prevents CAT from being packaged and distributed via PyPI, Bioconda or other systems. As the authors point out in the README, some operating systems (MacOS) may not be able to distinguish this from the posix "cat" command. My recommendation would be for the authors to choose a secondary or alternate entry point name themselves, to avoid confusion.

While CAT does not require many additional tools, it would benefit from either an "environment.yml" to create a matching conda environment, or just distribution via Bioconda. The latter is pretty easy to do given the simple structure of CAT, the only challenge being that a name for the entry point that can be added to "bin/" would have to be found.

I would also recommend that the authors decide whether cat is hosted at MGXlab/CAT or at dutilh/CAT. Currently, only the latter has the issue tracker enabled, leading me to believe it is the primary repository. There are tools for migrating the issues, if CAT is to be hosted at MGXlab. Otherwise, I would ask that the reference in the paper be adjusted to point to dutilh. To avoid confusion and to allow issue based interaction with the community.

Reviewer #2 (Remarks to the Author):

I find the changes to the manuscript to be a great improvement. I appreciate the not so small effort made to accommodate my requests. I hope the authors agree that the additional work made the study much stronger.

Reviewer #2 (Remarks on code availability):

The style of the code and the github repository are satisfactory. There is a decent README. The repo should make a release of v6.0 to correspond with the release of the manuscript. I did not have access to a sufficient computer to install and test run the code.

Reviewer #3 (Remarks to the Author):

All the concerns/issues I have raised in my first review, have now been properly addressed in the current revision and I thank the authors for the work. I haven't found any further issues.

Response to Reviewers

Reviewers' comments are shown in italic, authors' responses are not formatted.

Reviewer #1 (Remarks to the Author):

My thanks to the authors for addressing my concerns in full!

I have a few suggestions regarding the code distribution (see the new code section below). However, these are to be considered only as suggestions. I believe the manuscript is good for publication as is.

We thank Reviewer #1 for their constructive review and their time.

Reviewer #1 (Remarks on code availability):

CAT is implemented in Python as a collection of scripts with each pipeline step implemented in a single script, one file for shared code and an entry point script "CAT" used to dispatch commands. It is valid to do this, staying deliberately simple.

The choice of name for the entry point, "CAT", prevents CAT from being packaged and distributed via PyPI, Bioconda or other systems. As the authors point out in the README, some operating systems (MacOS) may not be able to distinguish this from the posix "cat" command. My recommendation would be for the authors to choose a secondary or alternate entry point name themselves, to avoid confusion.

We agree and have renamed the entry script to CAT_pack.

While CAT does not require many additional tools, it would benefit from either an "environment.yaml" to create a matching conda environment, or just distribution via Bioconda. The latter is pretty easy to do given the simple structure of CAT, the only challenge being that a name for the entry point that can be added to "bin/" would have to be found.

We have included the environment used for this manuscript as cat_pack_dependencies.yaml in a Zenodo repository (10.5281/zenodo.10731870). With the public release of CAT_pack v6.0 we will also distribute a conda package that will be kept up-to-date with the latest code. We are currently in the process of releasing the official CAT_pack v6.0 (it will be released on the 09th of March 2024).

I would also recommend that the authors decide whether cat is hosted at MGXlab/CAT or at dutilh/CAT. Currently, only the latter has the issue tracker enabled, leading me to believe it is the primary repository. There are tools for migrating the issues, if CAT is to be hosted at MGXlab. Otherwise, I would ask that the reference in the paper be adjusted to point to dutilh. To avoid confusion and to allow issue based interaction with the community.

We thank the reviewer for pointing this out. Currently MGXlab/CAT and dutilh/CAT are kept in sync and both are maintained, where the MGXlab version is a fork of dutilh. With the public release of CAT_pack v6.0, the main repository and its issue tracker will be moved to MGXlab and dutilh will be phased out.

Reviewer #2 (Remarks to the Author):

I find the changes to the manuscript to be a great improvement. I appreciate the not so small effort made to accommodate my requests. I hope the authors agree that the additional work made the study much stronger.

Reviewer #2 (Remarks on code availability):

The style of the code and the github repository are satisfactory. There is a decent README. The repo should make a release of v6.0 to correspond with the release of the manuscript. I did not have access to a sufficient computer to install and test run the code.

We thank Reviewer #2 for their appreciation of the revisions. We wholeheartedly agree that the study has become a lot stronger thanks to the review process.

We are currently in the process of releasing the official CAT_pack v6.0 (it will be released on the 09th of March 2024).

Reviewer #3 (Remarks to the Author):

All the concerns/issues I have raised in my first review, have now been properly addressed in the current revision and I thank the authors for the work. I haven't found any further issues.

We thank Reviewer #3 for their comments and the time invested in our manuscript.